# Fatal Neurodissemination and SARS-CoV-2 Tropism in K18-hACE2 Mice Is Only Partially Dependent on hACE2 Expression

**DOI:** 10.3390/v14030535

**Published:** 2022-03-05

**Authors:** Mariano Carossino, Devin Kenney, Aoife K. O’Connell, Paige Montanaro, Anna E. Tseng, Hans P. Gertje, Kyle A. Grosz, Maria Ericsson, Bertrand R. Huber, Susanna A. Kurnick, Saravanan Subramaniam, Thomas A. Kirkland, Joel R. Walker, Kevin P. Francis, Alexander D. Klose, Neal Paragas, Markus Bosmann, Mohsan Saeed, Udeni B. R. Balasuriya, Florian Douam, Nicholas A. Crossland

**Affiliations:** 1Louisiana Animal Disease Diagnostic Laboratory (LADDL), Louisiana State University, Baton Rouge, LA 61329, USA; mcarossino1@lsu.edu (M.C.); balasuriya1@lsu.edu (U.B.R.B.); 2Department of Pathobiological Sciences, School of Veterinary Medicine, Louisiana State University, Baton Rouge, LA 61329, USA; 3National Emerging Infectious Diseases Laboratories (NEIDL), Boston University, Boston, MA 02118, USA; kenneydj@bu.edu (D.K.); aocon@bu.edu (A.K.O.); pmontana@bu.edu (P.M.); aetseng@bu.edu (A.E.T.); hgertje@bu.edu (H.P.G.); kgrosz@bu.edu (K.A.G.); skurnick@bu.edu (S.A.K.); mbosmann@bu.edu (M.B.); msaeed@bu.edu (M.S.); 4Department of Microbiology, Boston University School of Medicine, Boston, MA 02118, USA; 5Department of Pathology and Laboratory Medicine, Boston University School of Medicine, Boston, MA 02118, USA; 6Electron Microscopy Core Facility, Harvard Medical School, Boston, MA 02115, USA; maria_ericsson@hms.harvard.edu; 7Department of Neurology, Boston University School of Medicine, Boston, MA 02118, USA; huberb@bu.edu; 8Department of Medicine, Pulmonary Center, Boston University School of Medicine, Boston, MA 02118, USA; ssubra@bu.edu; 9Promega Biosciences, LLC, San Luis Obispo, CA 93401, USA; thomas.kirkland@promega.com (T.A.K.); joel.walker@promega.com (J.R.W.); 10Perkin Elmer, Hopkinton, MA 01748, USA; kevin.francis@perkinelmer.com; 11InVivo Analytics Inc., New York, NY 10023, USA; klose@invivoax.com (A.D.K.); paragas@invivoax.com (N.P.); 12Department of Radiology Imaging Research Lab, University of Washington, Seattle, WA 98133, USA; 13Center for Thrombosis and Hemostasis, University Medical Center of the Johannes Gutenberg-University Mainz, 55131 Mainz, Germany; 14Department of Biochemistry, Boston University, Boston, MA 02118, USA

**Keywords:** translational animal model, comparative pathology, immunohistochemistry, in situ hybridization, viral pathogenesis, transmission electron microscopy, in vivo imaging

## Abstract

Animal models recapitulating COVID-19 are critical to enhance our understanding of SARS-CoV-2 pathogenesis. Intranasally inoculated transgenic mice expressing human angiotensin-converting enzyme 2 under the cytokeratin 18 promoter (K18-hACE2) represent a lethal model of SARS-CoV-2 infection. We evaluated the clinical and virological dynamics of SARS-CoV-2 using two intranasal doses (10^4^ and 10^6^ PFUs), with a detailed spatiotemporal pathologic analysis of the 10^6^ dose cohort. Despite generally mild-to-moderate pneumonia, clinical decline resulting in euthanasia or death was commonly associated with hypothermia and viral neurodissemination independent of inoculation dose. Neuroinvasion was first observed at 4 days post-infection, initially restricted to the olfactory bulb suggesting axonal transport via the olfactory neuroepithelium as the earliest portal of entry. Absence of viremia suggests neuroinvasion occurs independently of transport across the blood-brain barrier. SARS-CoV-2 tropism was neither restricted to ACE2-expressing cells (e.g., AT1 pneumocytes), nor inclusive of some ACE2-positive cell lineages (e.g., bronchiolar epithelium and brain vasculature). Absence of detectable ACE2 protein expression in neurons but overexpression in neuroepithelium suggest this as the most likely portal of neuroinvasion, with subsequent ACE2 independent lethal neurodissemination. A paucity of epidemiological data and contradicting evidence for neuroinvasion and neurodissemination in humans call into question the translational relevance of this model.

## 1. Introduction

The world is experiencing the devastating effects of the Coronavirus Disease 2019 (COVID-19) pandemic, a highly contagious viral respiratory disease caused by the newly emerged betacoronavirus, Severe Acute Respiratory Syndrome Coronavirus-2 (SARS-CoV-2) [1,2,3]. The initial index case was reported at a seafood market in Wuhan, Hubei Province, China in late 2019 [1]. While still under investigation, it has been postulated that the progenitor of SARS-CoV-2 may have originated from horseshoe bats (*Rhinolophus affinis*) or Malayan pangolins (*Manis javanica*) that, following spill over into humans, acquired the genomic features leading to adaptation and human-to-human transmission [1]. SARS-CoV-2 has a high transmissibility rate, and, to date, it has infected nearly 418 million people, resulting in over 5.85 million fatalities (17 February 2022) [4]. COVID-19 causes respiratory disease of variable severity, ranging from mild to severe, with the development of acute respiratory distress syndrome (ARDS) requiring intensive care and mechanical ventilation in a small fraction of patients [3,5,6,7]. Numerous comorbidities including hypertension, obesity, and diabetes, among others, are affiliated with an increased risk of developing severe COVID-19 [5,6,8,9,10]. Furthermore, a proportion of infected patients go on to develop poorly understood neurological signs and/or symptoms mostly associated with the loss of smell and taste (anosmia and ageusia), headache, dizziness, encephalopathy (delirium), cognitive decline and ischemic injury (stroke), in addition to a range of less common symptoms [5,7,11,12,13,14,15,16,17]. COVID-19 has severely challenged health care systems around the globe, with the urgent need for medical countermeasures including the development of efficacious vaccines and therapeutics especially with the continued emergence of variants of concern (VOC).

Animal models permissive to SARS-CoV-2 that serve as suitable models to help better understand the pathogenesis of COVID-19, while simultaneously assisting in the development and evaluation of novel vaccines and therapeutics to combat this disease, are critically needed [18,19,20]. While various animal models (mice, hamsters, non-human primates, ferrets, minks, dogs, and cats) have been evaluated to date [20,21,22,23,24,25,26,27,28], none faithfully recapitulates all the pathological features of COVID-19. The main limitation in the development of suitable murine models of COVID-19 is related to the virus entry mechanism: SARS-CoV-2 binds to target cells via interaction between the viral spike protein (S) and the host angiotensin-converting enzyme 2 (ACE2), considered to be the major host entry receptor [29]. The low binding affinity between the S protein and murine ACE2 (mACE2) renders conventional mouse strains naturally resistant to infection, posing a challenge in the development of murine models of COVID-19 [29,30,31,32]. These difficulties have been circumvented by the development of transgenic murine models that express human ACE2 (hACE2) under different promoters including hepatocyte nuclear factor-3/forkhead homologue 4 (HFH4), and cytokeratin 18 (K18) [28,33,34,35,36]. The transgenic murine model expressing hACE2 under a K18 promoter (namely K18-hACE2) was developed by McCray et al. in 2007 to study SARS-CoV [34], which shares the same host receptor as SARS-CoV-2 [37]. More recently, it has been shown that wild-type mice are permissive to the SARS-CoV-2 B.1.351 variant, which is attributed to a mutation in the Spike receptor-binding domain (RBD) resulting in high binding affinity to mACE2 [38].

SARS-CoV-2 infection of K18-hACE2 mice using wild-type, Alpha, and Beta variants results in lethal disease across a broad range of doses, analogous to that reported for SARS-CoV, with enhanced survival observed in Delta and Omicron variants, with the latter notably lacking evidence of neuroinvasion [28,34,36,39,40]. Early reports communicated lethality to be associated primarily with severe lung inflammation and impaired respiratory function, suggesting that the K18-hACE2 model recapitulates features of the respiratory disease observed in severe cases of COVID-19 diagnosed with ARDS [28,36]. However, the confounding impact of neuroinvasion and neurodissemination and its role in the clinical decline of SARS-CoV-2 infected K18-hACE2 mice is becoming more readily acknowledged [41,42,43,44]. Furthermore, administration of aerosolized SARS-CoV-2 to K18-hACE2 mice resulted solely in respiratory infection and limited clinical disease at 6 days post infection (6 dpi) in contrast with those inoculated intranasally, suggesting that severe clinical outcome in K18-hACE2 mice is attributed to CNS disease [45].

Under K18 regulation, gene expression analysis demonstrated *hACE2* to be expressed mainly in the lung and intestines, to a lower degree within kidney, liver, spleen, and small intestine, and to a relatively minor level of expression in the brain [34]. However, the cellular distribution of ACE2, and particularly hACE2, in tissues of K18-hACE2 mice remains largely undetermined. While this mouse model has been well-characterized, several gaps in our understanding of SARS-CoV-2 pathogenesis in this model remain, including the cellular dynamics of the inflammatory response in the lungs and neurodissemination, microscopic and ultrastructural features of neuronal damage, and spatial contextualization of hACE2 expression and viral tropism. We hypothesized that the nature, severity, and outcome of disease in the K18-hACE2 mouse model is not solely dictated by the expression and tissue distribution of hACE2 and that increased lethality in this model is ultimately related to severe neurodissemination, in part driven by regional ACE2 overexpression in the olfactory neuroepithelium (ONE) promoting neuroinvasion through axonal transit of the olfactory nerve. To investigate this hypothesis, we undertook a comprehensive spatiotemporal analysis of histologic and ultrastructural changes, cellular distribution of viral protein and RNA, viral loads, and titers, along with a detailed analysis of the distribution of *hACE2* mRNA, ACE2 protein, and its correlation with SARS-CoV-2 tropism as it pertains to this model. Thus, this study is the only to date that combines the use of classical virological approaches along with state-of-the-art quantitative spatial pathology tools to fill this critical gap.

Although SARS-CoV-2 protein and RNA were detected in ACE2-expressing cells such as epithelium of the olfactory mucosa and alveolar type 2 (AT2) cells, we found that SARS-CoV-2 USA-WA1/2020 primarily infected neurons and alveolar type 1 (AT1) pneumocytes, which lacked detectable ACE2 protein. Our results support neuroinvasion and subsequent near global neurodissemination within the brain as the primary cause of mortality in the K18-hACE2 mouse model irrespective of inoculation dose. These claims are supported by the observation that viral load and titers peaked in the brain when animals began to meet euthanasia criteria or succumb to disease. Clinically, this was reflected by onset of profound hypothermia and onset of neurological signs including tremors, stupor, and ataxia. Neurons within the brain in terminal animals displayed prominent spongiotic degeneration and necrosis, with concurrent detection of abundant viral protein, RNA, virally induced membrane modifications, and assembly of virus particles. Although pneumonia was uniformly observed and peaked at the time of death/euthanasia, it was of mild-to-moderate severity with declining viral loads in contrast with peak viral titers in the brain. Furthermore, several histologic hallmarks of severe COVID-19 were lacking in this model, (i.e., lack of diffuse alveolar damage and capillary microthrombi), suggesting pneumonia plays a contributing rather than a primary role as determinant of lethality in this model. The absence of detectable ACE2 protein and minimal *hACE2* mRNA in neurons, where viral particles, protein, and RNA were exclusively detected within the nervous system, suggests that viral neurodissemination is likely occurring via an ACE2 independent mechanism, with initial neuroinvasion occurring through ACE2 expressing neuroepithelium of the nasal passages. This study expands the current knowledge on the K18-hACE2 murine model better defining its strengths and limitations as a translational pre-clinical model for studying COVID-19.

## 2. Materials and Methods

**Biosafety.** All aspects of this study were approved by the Institutional Biosafety Committee and the office of Environmental Health and Safety at Boston University prior to study initiation. Work with SARS-CoV-2 was performed in a biosafety level-3 laboratory by personnel equipped with powered air-purifying respirators.

**Cells.** African green monkey kidney Vero E6 cells (ATCC^®^ CRL-1586™, American Type Culture Collection, Manassas, VA, USA) were maintained in Dulbecco’s minimum essential medium (DMEM; Gibco, Carlsbad, CA, USA [#11995-065]) containing 10% fetal bovine serum (FBS, ThermoFisher Scientific, Waltham, MA, USA), 1× non-essential amino acids (ThermoFisher Scientific), penicillin and streptomycin (100 U/mL and 100 μg/mL), and 0.25 μg/mL of amphotericin B (Gibco^®^, Carlsbad, CA, USA), and incubated at 37 °C and 5% CO_2_ in a humidified incubator.

**SARS-CoV-2 isolate stock preparation and titration.** All replication-competent SARS-CoV-2 experiments were performed in a biosafety level 3 laboratory (BSL-3) at the Boston University’ National Emerging Infectious Diseases Laboratories. 2019-nCoV/USA-WA1/2020 isolate (NCBI accession number: MN985325) of SARS-CoV-2 was obtained from the Centers for Disease Control and Prevention (Atlanta, GA, USA) and BEI Resources (Manassas, VA, USA). A passage 2 (P2) of SARS-CoV-2 was generated in Vero E6 cells by infecting these with the passage 1 at a multiplicity of infection (MOI) of 0.1. Cell culture media was harvested at 2 and 3 dpi and ultracentrifuged over a 20% sucrose cushion (Sigma-Aldrich, St. Louis, MO, USA). Pellets were then resuspended overnight at 4 °C in 500 µL of 1× PBS and titrated by plaque assay in Vero E6 cells following standard procedures. The titer of our P2 virus stock was 4 × 10^8^ plaque forming units (PFU)/mL.

**Recombinant SARS-CoV-2 NanoLuciferase stock.** Recombinant SARS-CoV-2 virus expressing a NanoLuciferase reporter (rSARS-CoV-2 NL) [46] was generously provided by the Laboratory of Dr. Pei-Yong Shi, University of Texas Medical Branch (UTMB). A passage 1 stock was produced as described above and similarly titrated in Vero E6 cells.

**Mice.** Mice were maintained in a facility accredited by the Association for the Assessment and Accreditation of Laboratory Animal Care (AAALAC). All protocols were approved by the Boston University Institutional Animal Care and Use Committee (PROTO202000020). Heterozygous K18-hACE2 C57BL/6J mice of both sexes (strain: 2B6.Cg-Tg(K18-ACE2)2Prlmn/J) were obtained from the Jackson Laboratory (Jax, Bar Harbor, ME). Animals were group-housed by sex in Tecniplast green line individually ventilated cages (Tecniplast, Buguggiate, Italy). Mice were maintained on a 12:12 light cycle at 30–70% humidity and provided ad-libitum water and standard chow diets (LabDiet, St. Louis, MO, USA). C57BL/6J mice (Jackson Laboratory) were used as experimental non-transgenic (background) controls when analyzing hACE2 (see below).

**Intranasal inoculation with SARS-CoV-2.** At 12–16 weeks of age, K18-hACE2 mice of both sexes were intranasally inoculated with either 10^4^ PFU or 10^6^ PFU of SARS-CoV-2 in 50 µL of sterile 1× PBS (n = 50 total [n = 28 male and n = 22 female], or sham inoculated with 50 µL of sterile 1× PBS (n = 6; 3 female and 3 male) as a means of establishing a survival curve. For pathology analysis, twenty-three animals inoculated with 1 × 10^6^ PFU were examined (n = 15 male and n = 8 female), which included two male survivors from the survival curve. Furthermore, an additional three female Sham/PBS inoculated negative controls (n = 3) were included for a total of 26 mice. These were examined at 2 dpi (n = 3), 4 dpi (n = 5), terminal endpoint (6–8 dpi, n = 13) and 14 dpi (survivors, n = 2), acknowledging we aimed to characterize severe CNS phenotypes. For virological analysis, additional animals were utilized specifically to harvest lung, brain, and serum samples at 2, 4 and 7 dpi. Sample numbers are outlined in the legend of Figure 1.

**Clinical monitoring.** Animals included in the 14-day survival curve studies mentioned above were intraperitoneally implanted with a radio frequency identification (RFID) temperature-monitoring microchip (Unified Information Devices, Lake Villa, IL, USA) 48–72 h prior to inoculation. An IACUC-approved clinical scoring system was utilized to monitor disease progression and establish humane endpoints (Table 1). Categories evaluated included body weight, general appearance, responsiveness, respiration, and neurological signs for a maximum score of 5. Animals were considered moribund and humanely euthanized in the event of the following: a score of 4 or greater for 2 consecutive observation periods, weight loss greater than or equal to 20%, severe respiratory distress, or lack of responsiveness. Clinical signs and body temperature were recorded once per day for the duration of the study. For design of the survival curve, animals euthanized on a given day were counted dead the day after. Animals found dead in cage were counted dead on the same day.

**In vivo 3D-imaging and analysis.** Before imaging, mice were administrated two, 75 μL subcutaneous injections of 1X PBS containing 0.65 uM Fluorofurimazine (FFz) substrate (Promega, Madison, WI, USA) for a total of 1.3 uM FFz per mouse. Mice were then imaged using a 3D-imaging mirror gantry isolation chamber (InVivo Analytics, New York, NY, USA) and an IVIS spectrum imager (PerkinElmer, Waltham, MA, USA). To perform imaging, mice were anesthetized with 2.5% isoflurane, placed into a body conforming animal mold (BCAM) (InVivo Analytics), and then imaged within 5 min of FFz injection. Images were acquired using a sequence imaging as followed; 60 s (s) open filter, 240 s 600 nm, 60 s open, 240 s 620 nm, 60 s open, 240 s 640 nm, 60 s open, 240 s 660 nm, 60 s open, 680 nm, 60 s open. Data analysis was performed using the cloud based In Vivo Plot software (In Vivo Analytics).

**Tissue processing and viral RNA isolation.** Twenty to 30 mg of tissue preserved in RNA*later* (Sigma-Aldrich, St. Louis, MO, USA; # R0901500ML) following harvest were placed into a 2 mL tube with 600 µL of RLT buffer with 1% β-mercaptoethanol and a 5 mm stainless steel bead (Qiagen, Valencia, CA, USA; #69989). Tissues were homogenized using a Qiagen TissueLyser II (Qiagen) by two dissociations cycles (two-minutes at 1800 oscillations/minute) with a one-minute rest in between. Samples were centrifuged at 17,000× *g* (13,000 rpm) for 10 min and supernatant was transferred to a new 1.5 mL tube. Viral RNA isolation was performed using a Qiagen RNeasy Plus Mini Kit (Qiagen; #74134), according to the manufacturer’s instructions. RNA was finally eluted in 30 μL of RNase/DNase-free water and stored at −80 °C until used.

**Quantification of infectious particles by plaque assay.** Between 20–40 mg of tissue (lung or brain) was homogenized using a Qiagen TissueLyser II (Qiagen) as described above in 500 μL of OptiMEM (ThermoFisher). Samples were clarified by centrifugation and ten-fold dilutions were inoculated in Vero E6 cells in a 12-well plate format and incubated at 37 °C for 1 h. After viral adsorption, cells were overlaid with 1 mL of a 1:1 mixture of 2× DMEM containing 4% FBS 1% penicillin/streptomycin and 2.4% Avicel (Dupont) and incubated for 3 days at 37 °C with 5% CO_2_. Cells were subsequently fixed in 10% formalin for 1 h, stained with 0.1% crystal violet in 10% ethanol/water for 1 h and washed with tap water. Viral titers were determined as PFU/mg of tissue.

**RNA isolation from serum.** Total viral RNA was isolated from serum using a Zymo Research Corporation Quick-RNA^TM^ Viral Kit (Zymo Research, Tustin, CA, USA; #R1040) according to the manufacturer’s instructions. RNA was eluted in 15 μL of RNase/DNase-free water and stored at −80 °C until used.

**SARS-CoV-2 E-specific reverse transcription quantitative polymerase chain reaction (RT-qPCR).** Viral RNA was quantitated using single-step RT-quantitative real-time PCR (Quanta qScript One-Step RT-qPCR Kit, QuantaBio, Beverly, MA, USA; VWR; #76047-082) with primers and TaqMan^®^ probes targeting the SARS-CoV-2 E gene as previously described^47^. The 20 μL reaction mixture contained 10 μL of Quanta qScript™ XLT One-Step RT-qPCR ToughMix, 0.5 μM of each primer E_Sarbeco_F1 and E_Sarbeco_R2, 0.25 μM of FAM-BHQ1 probe E_Sarbeco_P1, and 2 μL of template RNA. RT-qPCR was performed using an Applied Biosystems QuantStudio 3 (ThermoFisher Scientific) and the following cycling conditions: reverse transcription for 10 min at 55 °C, an activation step at 94 °C for 3 min followed by 45 cycles of denaturation at 94 °C for 15 s and combined annealing/extension at 58 °C for 30 s. For absolute quantitation of viral RNA, a 389 bp fragment from the SARS-CoV-2 E gene was cloned onto pIDTBlue plasmid under an SP6 promoter using NEB PCR cloning kit (New England Biosciences, Ipswich, MA, USA). The cloned fragment was then in vitro transcribed (mMessage mMachine SP6 transcription kit; ThermoFisher) to generate an RT-qPCR standard.

**Serum infectivity assay.** One day prior to the experiment, Vero E6 cells were plated into a 24-well plate, inoculated with 200 μL of OptiMEM containing 20 μL of serum or SARS-CoV-2 WA-isolate (MOI = 0.001 [positive control]), and incubated for 1 h at 37 °C. Media was subsequently removed and fresh DMEM containing 2% FBS and 1% penicillin/streptomycin was added. Cells were incubated at 37 °C with 5% CO_2_ for 48 h, 100 μL of supernatant was collected and RNA was extracted using a Zymo Research Corporation Quick-RNA^TM^ Viral Kit as per manufacturer’s instructions (Zymo Research) for analysis by RT-qPCR.

**Serum neutralization assay.** One day prior to the experiment, 1 × 10^4^ Vero E6 cells were plated into a 96-well plate. Serum was decomplemented at 56 °C for 30 min and an initial dilution of 1:10 was prepared in OptiMEM. Two-fold dilutions were subsequently prepared and mixed with rSARS-CoV-2 NL virus (MOI = 1) for 1 h at room temperature and then plated onto cells. After a 1-h incubation at 37 °C inoculum was removed and 200 μL of fresh DMEM containing 2% FBS and 1% penicillin/streptomycin was added. After a 24 h incubation at 37 °C with 5% CO_2_ media was removed and cells were fixed with 10% formalin for 1 h. A SARS-CoV-2 spike neutralizing antibody (Sino Biological Inc., Beijing, China; 2 μg/μL) was used as a positive control. Cells were washed with 1× PBS and 20 μM furimazine (MedChem Express, Monmouth Junction, NJ, USA) luciferin substrate was added onto cells. Cells were then imaged using an IVIS spectrum imager (PerkinElmer) and analyzed using LivingImage software (PerkinElmer). Titers were determined as the reciprocal of the highest dilution with >50% reduction of cytopathic effect.

**Histology.** Tissues from n = 26 mice were analyzed (Sham/PBS, n = 3; 2 dpi, n = 3; 4 dpi, n = 5; 6–8 dpi, n = 13; 14 dpi, n = 2). Lungs were insufflated with ~1.5 mL of 1% low melting point agarose (Sigma-Aldrich) diluted in 1× PBS using a 24-gauge catheter placed into the trachea. The skull cap was removed and the animal decapitated. Additional tissues harvested included the heart, kidneys, and representative sections of the gastrointestinal tract, which included the duodenum, jejunum, ileum, cecum, and colon. Tissues were inactivated in 10% neutral buffered formalin at a 20:1 fixative to tissue ratio for a minimum of 72 h before removal from BSL-3 in accordance with an approved institutional standard operating procedure. Following fixation, the whole head was decalcified in Immunocal™ Decalcifier (StatLab, McKinney, TX, USA) for 7 days before performing a mid-sagittal section. Tissues were subsequently processed, embedded in paraffin and five-micron sections stained with hematoxylin and eosin or Luxol Fast Blue (myelin stain) following standard histological procedures.

**Immunohistochemistry and RNAscope^®^ in situ hybridization.** Immunohistochemistry (IHC) was performed using a Ventana BenchMark Discovery Ultra autostainer (Roche Diagnostics, Indianapolis, IN, USA). Specific IHC assay details including antibodies, protein retrieval, sequence of multiplex assays, and incubation periods are found in Table 2. SARS-CoV-2 S was semiquantitatively scored as follows: 0, no viral protein observed; 1, up to 5% positive cells per 400× field examined; 2, 5–25% positive cells per 400× field examined; and 3, up to 50% positive cells per 400× field examined.

For SARS-CoV-2 RNAscope^®^ ISH, an anti-sense probe targeting the spike (S; nucleotide sequence: 21,563–25,384) of SARS-CoV-2, USA-WA1/2020 isolate (GenBank accession number MN985325.1) was used as previously described [47]. The RNAscope^®^ ISH assay was performed using the RNAscope 2.5 LSx Reagent Kit (Advanced Cell Diagnostics, Newark, CA, USA) on the automated BOND RXm platform (Leica Biosystems, Buffalo Grove, IL, USA) as described previously [23]. A SARS-CoV-2-infected Vero E6 cell pellet was used as a positive assay control. For all assays, an uninfected mouse was used as a negative control.

For *hACE2* mRNA RNAscope^®^ ISH, an anti-sense probe targeting *hACE2* (GenBank accession number NM_021804.3; Cat. No. 848038) with no cross-reactivity to murine *Ace2* was used in a similar manner as described above with the exception that AMP5 and AMP6 were incubated for 45 min and 30 min, respectively. Murine *peptidylprolyl isomerase B (Ppib)* mRNA was used as a housekeeping gene to determine RNA quality and a Vero E6 cell pellet was used as a positive assay control.

**Multispectral microscopy.** Fluorescently labeled slides were imaged using a Mantra 2.0^TM^ or Vectra Polaris^TM^ Quantitative Pathology Imaging System (Akoya Biosciences, Marlborough, MA, USA). To maximize signal-to-noise ratios, images were spectrally unmixed using a synthetic library specific for the Opal fluorophores used for each assay and for 4′,6-diamidino-2-phenylindole (DAPI). An unstained lung or brain section were used to create a tissue specific autofluorescence signature that was subsequently removed from whole-slide images using InForm software version 2.4.8 (Akoya Biosciences).

**Quantitative image analysis of multiplex immunohistochemistry.** Digitized whole slide scans were analyzed using the image analysis software HALO (Indica Labs, Inc., Corrales, NM, USA). Slides were manually annotated to include only the brain and/or lung parenchyma depending on the panel being evaluated. Visualization threshold values were adjusted in viewer settings to reduce background signal and fine-tune visibility of markers within each sample. For the CNS panel, area quantification (AQ) was performed to determine percentages of SARS-CoV-2 Spike, Iba1 (microglia) and GFAP (astrocyte) immunoreactivity. For the lung panel, we employed the HALO Highplex (HP) module. Individual cells were identified using DAPI to segment individual nuclei. Minimum cytoplasm and membrane thresholds were set for each dye to detect positive staining within a cell. Parameters were set using the real-time tuning mechanism that was tailored for each individual sample based on signal intensity. Phenotypes were determined by selecting inclusion and exclusion parameters relating to stains of interest. The algorithm produces a quantitative output for each cell phenotype as well as total cells per total area analyzed for an output of cells/µm^2^. The AQ module was also used the lung panel for quantification of SARS-CoV-2-Spike immunoreactivity.

**Quantitative image analysis of brightfield microscopy.** Digitized whole slide scans of hematoxylin and eosin (H&E) stained mouse lungs were analyzed using the Halo Tissue Classifier module. TC is a train-by-example machine learning algorithm used to identify dissimilar areas of tissue based on contextual features. For these lung samples, a classifier was created to distinguish areas of pneumonic lung from normal parenchyma. The classifier was run on whole lung images to determine the percentage of pneumonia. Quantitative outputs are given as total classified area (mm^2^), normal lung area (mm^2^), and pneumonia area (mm^2^). We divided pneumonic area by total classified area to generate a percentage of pneumonia for statistical analysis.

**Transmission electron microscopy (TEM).** Lung and brain samples from a single K18-hACE2 mouse infected with 10^6^ PFUs of SARS-CoV-2 that met euthanasia criteria at 6 dpi were fixed for 72 h in a mixture of 2.5% Glutaraldehyde and 2% formaldehyde in 0.1 M sodium cacodylate buffer (pH 7.4). Samples were then washed in 0.1M cacodylate buffer and postfixed with 1% Osmiumtetroxide (OsO4)/1.5% Potassiumferrocyanide (KFeCN6) for 1 h at room temperature. Samples were stained with 1% tannic acid (Electron Microscopy Sciences) in water for 1 h followed by 50 mM Maleate buffer pH 5.15 (MB) and incubated in 1% uranyl acetate in MB for 1 h. Following washes in MB and water, the samples were dehydrated in grades of alcohol, placed in propyleneoxide for 1 h and infiltrated ON in a 1:1 mixture of propyleneoxide and TAAB Epon. The following day the samples were embedded in fresh TAAB Epon and polymerized at 60 °C for 48 h. Semi-thin (0.5 μm) and ultrathin sections (50–80 nm) were cut on a Reichert Ultracut-S microtome (Leica). Semi-thin sections were stained with Toluidine blue for examination with a light microscope to find affected areas in the tissue. Ultrathin sections from those areas were picked up onto formvar/carbon coated copper grids, stained with 1% uranyl acetate in 50% acetone followed by 0.2% lead citrate and examined in a JEOL 1200EX transmission electron microscope (JEOL, Akishima, Tokyo, Japan). Images were recorded with an AMT 2k CCD camera.

**Statistical analysis.** Descriptive statistics and graphics as well as Kaplan-Meier (survival) curves and statistical tests were performed using GraphPad Prism v9.3.1 statistical analysis software (GraphPad, San Diego, CA, USA). Clinical parameters and quantitative pathology results were analyzed using either a two-way or one-way ANOVA with Dunnett post-hoc analysis with means of groups compared to the Sham/PBS-inoculated negative controls. Viral load data were evaluated using either a one-way (serum qPCR) or two-way ANOVA (tissue qPCR and PFU data) with Tukey post hoc analysis. Significance levels were set at *p*-value < 0.05 in all cases. Statistical significance on figures and Appendix A is labelled as follows: * *p* ≤ 0.05, ** *p* ≤ 0.01, *** *p* ≤ 0.001, **** *p* ≤ 0.0001.

## 3. Results

### 3.1. SARS-CoV-2 Is Nearly Invariably Fatal in Infected K18-hACE2 Mice over a 10^4^–10^6^ PFU Inoculation Dose

K18-hACE2 mice inoculated intranasally with SARS-CoV-2 (n = 50 [n = 28 male and n = 22 female]) began losing weight as early as 4 days post-infection (dpi) irrespective of dose (1 × 10^4^ or 1 × 10^6^ PFUs) and sex, with maximum weight loss occurring at 6–7 dpi (15.9 ± 1.1% in male mice, 19.7 ± 1.6% in female, and combined 17.8 ± 2.4%; Figure 1A). Trends in weight loss paralleled increasing clinical scores and declines in core body temperature, with the latter two precipitously increasing or decreasing, respectively, near the time of death (Figure 1B,C). In both doses, SARS-CoV-2-infected K18-hACE2 mice exhibited neurological signs starting 6 dpi, characterized by profound stupor, tremors, proprioceptive defects, and abnormal gait, with most animals euthanized or found dead in their cage by 8 dpi (~94%; 47/50 (Figure 1D)). At the time of death (6–7 dpi), the median clinical score ranged from 2.8 at 6 dpi (interquartile range = 0.083) to 4.2 at 7 dpi (interquartile range = 1.3) and the mean body temperature was 30.5 ± 1.5 °C. All three survivors were male mice (2 in the high dose and 1 in the low dose) and did not display hypothermia during the observation period, a feature that was consistently observed in animals that succumbed to disease or met euthanasia criteria. PBS/Sham-inoculated control male (n = 3) and female (n = 3) mice did not exhibit weight loss, clinical signs, or hypothermia throughout the course of the study.

Peak of lethality was associated with a significant increase in viral loads (both viral RNA copies and infectious virus particles) in the brain of the K18-hACE2 mice irrespective of inoculation dose (Figure 1E–G), as previously reported [36,42,43]. No lethality was recorded in sham-inoculated mice (n = 6). In the lung, viral RNA copies were detectable at the earliest experimental timepoint (2 dpi) and remained stable over time, consistently within the value range reported in previous studies [28]. While viral RNA remained high, viral titers however gradually declined over time highlighting that infectious viral particle production is rapidly controlled by host responses while infected cells may persist longer. In contrast, viral RNA, and infectious particles in the brain of both doses dramatically increased over time (Figure 1E–F). Quantification of infectious viral particles in the brain at 7 dpi represented the highest mean viral titer observed during the study. A small amount of viral RNA was detected in the serum (Figure 1H); however, incubation of SARS-CoV-2 permissive Vero E6 cells with serum samples did not result in any detectable productive infection in vitro, confirming an absence of viremia in intranasally-inoculated K18-hACE2 mice (Figure 1I). Altogether, our data illustrate that lethality was associated with increasing viral RNA loads and infectious virus particles in the brain, while simultaneously declined in the lung by the time of death or euthanasia.

### 3.2. SARS-CoV-2 Results in Transient Mild Infection in the Nasal Cavity of K18-hACE2 Mice

We next performed detailed histologic analysis of various tissues to uncover the morphologic correlates of lethality in K18-hACE2 mice. For this, we first focused on the spatial and temporal dynamics of SARS-CoV-2 infection in the upper respiratory tract and analyzed the anterior/rostral nasal cavity (Figure 2A–F) and olfactory neuroepithelium (Figure 2G–L) for disease-associated lesions, viral antigen, and RNA at 2 dpi (n = 3), 4 dpi (n = 5), 6–8 dpi-terminal (n = 13) and surviving mice at 14 dpi (n = 2).

At 2 dpi, the anterior/rostral nasal cavity was characterized by mild, multifocal neutrophilic inflammation (rhinitis) with segmental degeneration and necrosis of transitional and respiratory epithelium (Figure 2B), which colocalized with intracytoplasmic SARS-CoV-2 protein and RNA (Figure 2E). Adjacent nasal passages were partially filled with small amounts of cellular debris, degenerate neutrophils, and small numbers of erythrocytes. The lamina propria underlying affected areas was infiltrated by low to mild numbers of neutrophils and fewer lymphocytes (Figure 2B). At 4 dpi, epithelial degeneration and necrosis in the rostral and intermediate turbinates was no longer observed, replaced by mild residual lymphocytic rhinitis and rare neutrophils within the lamina propria (Figure 2C), and absence of exudate within nasal passages. SARS-CoV-2 protein and RNA were less commonly observed and restricted to rare positive cells in the respiratory epithelium (Figure 2F and Table 3 and Appendix A). By 6–8 dpi (terminal disease), the anterior/rostral nasal cavity was histologically within normal limits and no SARS-CoV-2 protein or RNA were detectable (Table 3 and Appendix A).

The posterior nasal cavity, olfactory neuroepithelium (ONE) (Figure 2G–L), displayed mild segmental degeneration and necrosis at 2 dpi, which colocalized with abundant SARS-CoV-2 protein and RNA (Figure 2K and Table 3 and Appendix A). By 4 dpi, histopathologic lesions in the ONE had resolved, but rare SARS-CoV-2 protein and RNA were observed both at 4 and 7 dpi (Figure 2L and Table 3). No SARS-CoV-2 protein or RNA were detected in the ONE by 14 dpi (Table 3 and Appendix A). No alterations were noted in PBS/Sham inoculated, control mice (n = 3).

### 3.3. SARS-CoV-2 Induces Moderate Interstitial Pneumonia in K18-hACE2 Mice

Lungs were histologically assessed at 2, 4, 6–8 dpi-terminal disease (n = 3, n = 5 and n = 13, respectively) and at 14 dpi for those mice who survived (n = 2). In the lower respiratory tract, histologic alterations in the pulmonary parenchyma mainly involved the alveoli, interstitium and perivascular compartments (Figure 3A–K). Overall, pathologic alterations in the lungs were characterized by mild-to-moderate progressive lymphohistiocytic and mild neutrophilic interstitial pneumonia that peaked at terminal disease (6–8 dpi) (Figure 3G,H). Quantitative histologic analysis performed on insufflated lungs from mice at each timepoint confirmed that peak disease occurred at 7 dpi, with a mean of ~10% of total lung area affected and only a single outlier with involvement of ~40% of total lung area, suggesting that more severe disease is possible, albeit uncommon (Figure 3K, Sham/PBS-infected mice served as controls). Of note, data from sub-optimally insufflated lungs were excluded from the classier, as the algorithm falsely labeled areas of atelectasis as pneumonia. Pneumonia was interpreted to be minimal to non-existent at 2 dpi (and in agreement to previous reports in this model) and, thus, considered negligible and also excluded [28]. Furthermore, we only included 7 dpi inoculated animals for the lung classifier as we had too few animals at the 6 and 8 dpi timepoints to make any meaningful conclusions. No alterations were noted in PBS/Sham-infected, control mice (n = 3).

At 2 dpi, minimal perivascular and peribronchiolar inflammation, consisting primarily of lymphocytes and histiocytes, and occasional perivascular edema were observed (Figure 3C). Pulmonary vessels were frequently reactive and lined by a plump endothelium with marginating leukocytes (Figure 3D). SARS-CoV-2 protein and RNA (Appendix A) were observed in proximity to areas of interstitial pneumonia and localized within the cytoplasm of alveolar type (AT) 1 (squamous epithelium) and fewer AT2 cells (cuboidal epithelium) (Appendix A).

At 4 dpi, peak in viral protein and RNA abundance were observed (correlating with the highest viral titer and RNA load as determined by RT-qPCR and plaque assays) (Figure 1E,F; Appendix A) along with increasing lymphohistiocytic and neutrophilic infiltrate (Figure 3E,F). SARS-CoV-2 cellular tropism did not differ from that described at 2 dpi (Appendix A).

At 7 dpi, lymphohistiocytic and neutrophilic interstitial pneumonia peaked in severity, which on average was moderate to regionally severe, affecting ~10–40% of the parenchyma (Figure 3G,H,K). Additional unique findings at 7 dpi included rare alveolar septal necrosis, mild proliferation of AT2 cells, and sporadic regional pulmonary edema (Figure 3G,H). SARS-CoV-2 protein and RNA were occasionally still abundant in several animals, but predominated in histologically normal parenchyma, with minimal to rare detection in areas of prominent inflammation (Appendix A and Table 3, and Appendix A).

In the two survivors euthanized at 14 dpi, persistent mild to moderate lymphohistiocytic interstitial pneumonia was observed, with formation of sporadic lymphoid aggregates and mild persistence of AT2 hyperplasia (Figure 3I,J). SARS-CoV-2 protein or RNA were no longer detectable at 14 dpi (Appendix A), but detection of neutralizing antibodies confirmed infection in these animals (see below).

Of note, no evidence of SARS-CoV-2 infection was observed in bronchiolar epithelium and pulmonary vasculature at any time during the study (Figure 3 and Appendix A and Table 3 and Appendix A). Similarly, hyaline membranes, vascular thrombosis, and syncytial cells were not observed at any time point across all animals, which contrasts with disease described in human autopsies [48] and non-human primate studies [49,50]. In one animal (7 dpi), there was localized flooding of bronchioles by degenerate neutrophils and cellular debris mixed with birefringent foreign material consistent with aspiration pneumonia, a rare complication previously reported in K18-hACE2 mice infected with SARS-CoV that was ultimately attributed to pharyngeal and laryngeal dysfunction impeding normal swallowing reflex, a sequela secondary to central nervous system (CNS) disease [34].

Altogether, our data displays evidence of a significant but generally mild-to-moderate lymphohistiocytic interstitial pneumonia in SARS-CoV-2 infected K18-hACE2 mice. Histopathological features contrast with those observed in severe cases of COVID-19 in humans and suggest that the lethality observed in this model is in part independent of virally induced lung injury and resultant pneumonia.

### 3.4. Pulmonary SARS-CoV-2 Replication and Assembly in K18-hACE2 Mice Occurs Exclusively in AT1 and AT2 Cells

Subsequently, we aimed to further investigate SARS-CoV-2 tropism in the lower respiratory tract of K18-hACE2 mice. We first performed qualitative multiplex IHC to probe the localization of SARS-CoV-2 protein in AT1 cells (cell marker: receptor for advanced glycation end-products (RAGE), AT2 cells (cell marker: surfactant protein C (SPC), and endothelial cells (cell marker: CD31) at 4 dpi. PBS/Sham-infected mice were used as controls. SARS-CoV-2 protein was restricted within RAGE+ AT1 and SPC+ AT2 pneumocytes, but not observed within CD31+ endothelial cells (Figure 4A–C). The pulmonary parenchyma was further evaluated ultrastructurally near the peak of viral replication (6 dpi) in an animal euthanized after having met euthanasia criteria with tissues specifically processed for transmission electron microscopy (TEM). Ultrastructurally, virus particles were exclusively observed bound by membrane bound vesicles in the cytoplasm of cells containing caveolae (AT1 cells) or lamellar bodies (AT2 cells) (Figure 4D–F, respectively). No virus particles or viral induced membrane modifications were observed in vascular endothelial, ciliated, or non-ciliated (Club cells) bronchiolar epithelial cells (Figure 4G–H). Of note, cubic membranes (CuMs) affiliated with virus particles was a distinctive feature rarely observed solely in AT1 pneumocytes (Figure 4E).

### 3.5. Effective Control of SARS-CoV-2 Infection in the Lower Respiratory Tract Is Associated with Recruitment of Macrophages and to a Lesser Degree Cytotoxic T Cells

Next, we quantitatively characterized the cell density of inflammatory cells (cells/μm^2^) including macrophages (Iba1), cytotoxic T cells (CD8), B cells (CD19) and total area immunoreactivity (% area μm^2^) of viral protein (Spike) in the lungs of SARS-CoV-2 infected K18-hACE2 mice (Figure 5A–H). SARS-CoV-2 S immunoreactivity peaked between 4–7 dpi (Figure 5A), supporting a positive correlation between viral infection and the progressive inflammatory cell infiltrate, but was not statistically significant across groups. We attribute this finding to our low sample size for quantitative whole slide analysis, and individual animal variability likely represented by the inherent heterogeneity of viral pneumonia. Iba1+ macrophages represented the predominant inflammatory infiltrate across all time points with a temporal increase peaking at 7 dpi (*p* = 0.0044 compared to sham inoculated mice, Figure 5B,G). Cytotoxic T cells were the second most abundant inflammatory infiltrate quantified, which also displayed a temporal increase peaking around 4–7 dpi (Figure 5C,F,G); however, these cells were present at a ~10-fold reduced frequency compared to macrophages and statistical significance was not observed across timepoints, suggesting an early and plateaued response of this inflammatory population. B cells were elevated by 7 dpi but reached peak cell density at 14 dpi (Figure 5D,H), the only time point where discrete lymphoid aggregates were observed histologically (*p* ≤ 0.0001 compared to sham inoculated mice). Altogether, our data suggests that a strong and persistent macrophage infiltration and, to a lesser degree, infiltrating cytotoxic T cells are important contributors to the decline of viral load that occurs in the lungs between 4–7 dpi, with B cells potentially being involved if animals survive the acute stage of disease. Of note, neutralizing antibodies to SARS-CoV-2 were confirmed in the two survivor mice (Figure 5H,I). Lungs from PBS/Sham-infected mice served as baseline controls for quantitative analysis.

### 3.6. SARS-CoV-2 Exhibits Extensive Neuroinvasion with Resultant Neuronal Degeneration and Necrosis in K18-hACE2 Mice

Pursuing our hypothesis that the lethality of the K18-hACE2 mice is associated with neurodissemination, we analyzed sagittal sections of the whole head to characterize distribution of viral protein and RNA and progression of histologic lesions at different timepoints post-infection (2, 4, terminal disease 6–8 dpi and 14 dpi).

Temporal distribution of viral antigen and RNA in the brain is shown in Figure 6A. First detectable within mitral and inner nuclear neurons of the olfactory bulb and small clusters of neurons within the anterior olfactory nucleus and orbital area of the cerebral cortex at 4 dpi, SARS-CoV-2 protein and RNA had a widespread distribution throughout the brain in roughly 85% (11/13) of infected K18-hACE2 that reached euthanasia criteria between 6–8 dpi. Viral antigen and RNA was exclusively identified within neuronal cell bodies and processes, including neuronal bodies within the cerebral cortex, CA1, CA2 and CA3 regions of the hippocampus, anterior olfactory nucleus, caudoputamen, nucleus accumbens, thalamic nuclei including hypothalamus, midbrain, pons and medulla oblongata nuclei (Figure 6A). Few vestibulocochlear nerve fascicles and retinal ganglion cells showed immunoreactivity for viral protein (Appendix A). Neuronal morphologic changes directly corresponded with abundance of SARS-CoV-2 S protein and viral RNA, with severe and widespread alterations in the brain in animals reaching euthanasia criteria between 6–8 dpi (n = 13, Figure 7A,B), including the olfactory bulb, cerebral cortex (most predominantly somatosensory and somatomotor areas), hippocampus (mainly CA1 region), midbrain (thalamus and hypothalamus), brainstem, and the dentate nucleus. Affected neuroparenchyma exhibited moderate-to-regionally marked neuronal spongiosis with loss of Nissl substance/chromatolysis and multifocal shrunken, angular, hypereosinophilic and pyknotic neuronal bodies (neuronal degeneration and necrosis, Figure 7B [insets]) occasionally delimited by multiple glial cells (satellitosis), diffuse reactive gliosis adjacent to areas of neuronal degeneration and necrosis, and mild sporadic delicate lymphocytic perivascular cuffing. Notably, the cerebellum (cortical layers and associated white matter of the cerebellar folia) was spared of histologic changes. Neurodissemination was also further confirmed temporally using a NanoLuc expressing recombinant SARS-CoV-2 virus (Rsars-CoV-2 NL), with increased bioluminescence in the brain and lower signal in the lungs of a representative K18-hACE2 mouse at 6 dpi (Figure 6B).

We quantitatively examined the glial response in infected K18-hACE2 at 2 dpi (n = 3), 4 dpi (n = 4), and 7 dpi (n = 5) and compared it to PBS/Sham-infected mice (n = 3; Figure 7C–E). The total immunoreactive area (%) for astrocytes (GFAP+) and microglia (Iba-1+) dramatically increased (astrogliosis and microgliosis) at 7 dpi compared to sham inoculated negative controls (GFAP, *p* = 0.0101; Iba1, *p* = 0.0327), and corresponded to the peak expression of SARS-CoV-2 S protein, which was also significantly increased compared to PBS/Sham inoculated negative controls (*p* = 0.0351; Figure 7C–E). Morphologically, astrocytic processes at this terminal timepoint were broad with extensive branching compared to PBS/Sham-inoculated animals and those evaluated at 2 and 4 dpi. Similarly, microglial cytoplasmic processes were notably broad and shortened compared to 2 dpi, 4 dpi, and PBS/Sham inoculated negative controls (Figure 7C,D). Temporally linked reactive microgliosis and astrogliosis with peak neurodissemination, suggests that activation of these cells could contribute to neuronal injury either through direct neurotoxic and/or loss of normal homeostatic neurotrophic mechanisms that warrant future research.

Dramatic ultrastructural changes were noted in affected areas of the cerebral cortex and hippocampus of an animal euthanized after meeting criteria at 6 dpi, which included the presence of neurologic signs (tremors and ataxia). The histologic phenotype of neuronal spongiosis and loss of Nissl substance/chromatolysis corresponded to neuronal bodies containing numerous virus particles (VPs: mean diameter of 94.5 nm; standard deviation of 13.5 nm; n = 62) bound by membrane bound vesicles and viral induced membrane modifications (Figure 8A–H). The latter included viral replication organelles (Ros) such as those previously described in a diverse array of coronaviruses [51]. These included double-membrane vesicles (DMVs: mean diameter of 429.1 nm; standard deviation of 96.8 nm; n = 60), double-membrane spherules (DMSs: mean diameter of 80.1 nm; standard deviation of 5.1 nm; n = 38) and convoluted membranes (CMs), all of which were observed exclusively in the cytoplasm of neurons, with no detection in glial cells, endothelium or pericytes. Neuronal pyknosis and karyolysis were represented by global electron dense transformation of the cytoplasm (neuronal necrosis) in the presence of residual viral induced membrane modifications (Figure 8A–H).

Considering the severe bladder distention noted at necropsy and proprioceptive deficits observed clinically, we examined the cervicothoracic and lumbosacral segments of the spinal cord. In 9/11 animals that died or were euthanized due to terminal disease, similar histologic findings were observed as those described in the brain, albeit with milder gliosis and lymphocytic perivascular cuffing (Appendix A). We also observed mild-to-moderate detection of viral protein in the spinal cord that predominated within neurons of the cervicothoracic segments (Appendix A, and Table 3 and Appendix A). Finally, Luxol Fast Blue was utilized to visualize the integrity of myelin following SARS-CoV-2 invasion in the brain and spinal cord at 7 dpi, with no evidence of demyelination noted (Appendix A).

Our data illustrates that SARS-CoV-2 infection of K18-hACE2 results in severe neuronal invasion of the CNS, via transport to the olfactory bulb originating from axonal processes traversing the ONE. Viral neuroinvasion resulted in extensive neuronal cytopathic effect in infected cells that ultimately resulted in cell death. Further research is warranted to characterize the role of uninfected but reactive microglia and astrocytes in SARS-CoV-2 neuronal injury using a multidimensional approach including molecular and functional testing.

### 3.7. ACE2 Expression and Distribution Does Not Fully Reflect SARS-CoV-2 Cellular Tropism in K18-hACE2 Mice

To further explore the mechanism driving lethal SARS-CoV-2 infection in K18-hACE2 mice, we investigated the tissue and cellular distribution of the ACE2 receptor and *hACE2* mRNA in PBS/Sham as well as SARS-CoV-2-infected K18-hACE2 mice and non-transgenic wild-type (WT) C57BL/6J mice (Figure 9A–L). For IHC, we utilized a cross-reactive anti-ACE2 antibody (cross-reactive to hACE2 and mACE2) (Table 2). In the lower respiratory tract (lungs), ACE2 was ubiquitously expressed along the apical membrane of bronchiolar epithelium and, less commonly, in rare and scattered AT2 pneumocytes (Figure 9A–C). No ACE2 expression was found in AT1 pneumocytes. No evident differences in the distribution and abundance of ACE2 expression were identified between C57BL/6J, PBS/Sham-inoculated K18-hACE2, and terminal SARS-CoV-2 inoculated K18hACE2 mice (7 dpi). These findings were further confirmed by analyzing expression and distribution of *hACE2* mRNA using RNAscope^®^ ISH (Figure 10). Although no expression of *hACE2* mRNA was detected in the lungs of non-transgenic WT C57BL/6J mice (Figure 10A), expression of *hACE2* mRNA was detectable, but of low expression in the lungs of K18-hACE2 mice, and mostly involved bronchiolar epithelial cells with sporadic expression in AT2 pneumocytes (Figure 10B,C). These findings therefore suggest that *hACE2* expression might not be the sole host factor determinant of susceptibility to SARS-CoV-2. This is clearly exemplified by the following: (1) certain cell types that, while expressing *hACE2*, were non-permissive to SARS-CoV-2 infection throughout the experiment (i.e., bronchiolar epithelial cells); and (2) the near diffuse infection of AT1 cells by 4 dpi despite absent expression of *hACE2* in these cells. Altogether, these observations then support evidence for an ACE2-independent viral entry mechanism playing a major role in the pulmonary dissemination of SARS-CoV-2 in K18-hACE2 mice.

In contrast to the lung, ACE2 protein was clearly overexpressed in the nasal cavity of K18-hACE2 mice compared to C57BL/6J mice. We assessed ACE2 protein expression on the rostral transitional epithelium, respiratory epithelium at the level of the intermediate turbinates, as well as in the ONE and olfactory bulb (Figure 9D–F,G–I). In contrast with WT C57BL/6J mice, in which ACE2 was undetectable within the nasal cavity, ACE2 protein was diffusely expressed within the apical membrane of transitional and respiratory epithelium, and segmentally within the apical surface of the ONE in both sham-inoculated and SARS-CoV-2-infected K18-hACE mice (Figure 9D–F). For the ONE and respiratory epithelium of rostral turbinates, estimation of *hACE2* abundance and distribution could not be accurately assessed since the decalcification procedure is believed to have had a significant impact in the quality of cellular mRNA as demonstrated by the low detection of the housekeeping mRNA, *Ppib*.

In the brain of both non-transgenic WT C57BL/6J and K18-hACE2 mice, ACE2 protein was observed in blood vessels (Figure 9J–L), as well as ependymal and choroid plexus epithelium. In contrast, *hACE2* mRNA expression was overall low and its distribution involved clusters of neurons within the cerebral cortex, hippocampus, midbrain, brainstem, and Purkinje cells from the cerebellum, with no expression noted in non-transgenic WT C57BL/6J mice (Figure 10D–F). No vascular expression of *hACE2* mRNA was observed in the brain. Taken together, our data show a discrepancy between ACE2 protein and RNA expression and distribution within the CNS. This is partly attributable to the fact that the ACE2 antibody we utilized cross reacts with both hACE2 and mACE2 proteins, while the *ACE2* probe employed was human specific. The absence of *hACE2* hybridization with simultaneous ACE2 immunoreactivity in the CNS blood vessels supports the notion that ACE2 expression in these cells is of murine origin. The absence of ACE2 immunoreactivity in neurons is suggestive of a potential restriction in the translation (or post-translation) of the ACE2 protein in these cells. This, in addition to the fact that Purkinje cells of the cerebellum do not appear permissive to SARS-CoV-2 infection despite the low expression of *hACE2* mRNA, suggests that ACE2 is likely not the sole host factor associated with neuroinvasion and that other ACE2-independent entry mechanisms contribute to neuroinvasion and spread by SARS-CoV-2 in this murine model. Alternatively, and/or in parallel, the overexpression of ACE2 protein within the nasal passages may be sufficient to enhance neuroinvasion by enhancing axonal transport via the ONE.

### 3.8. Absence of Infection and Histologic Lesions in Extrapulmonary and Extraneural Tissues Despite ACE2 Expression

Other tissues examined included heart, kidney, stomach, duodenum, jejunum, ileum, cecum, and colon. All of these were histologically within normal limits and no SARS-CoV-2 S protein was detected in any of these tissues at any time point (Table 1). ACE2 distribution was evaluated in sections of the heart, stomach, small intestine, and colon. While ACE2 expression was limited to the blood vessels in the heart and glandular stomach, intense expression was noted in the non-glandular mucosa of the stomach (Appendix A) and apical surface of enterocytes lining the small intestinal mucosa (Appendix A). Colonic enterocytes sporadically expressed ACE2 (Appendix A).

## 4. Discussion

The K18-hACE2 transgenic mouse model has become a widespread laboratory animal model suitable for studying SARS-CoV-2 pathogenesis as well as evaluating efficacy of medical countermeasures against COVID-19 [18]. Except for the Omicron variant (which lacks neuroinvasion and induces minimal lung disease), K18-hACE2 mice develop lethal disease associated with mild-to-moderate pulmonary pathology and lethal neurodissemination, corroborated by quantification of peak viral loads in the brain during terminal disease [28,34,35,36,41,52,53,54]. In contrast, several other adult murine models of SARS-CoV-2 (e.g., adenovirus-transduced hACE2 mice and hACE2 knock-in mice, as well as the use of mouse-adapted SARS-CoV-2 strains in wild-type mice) develop only mild pulmonary disease with limited and transient viral replication, and low to no lethality [32,35,55,56]. While the K18-hACE2 murine model has been informative in shedding light on mechanisms of lung injury and dysfunction, it fails to faithfully recapitulate several key histologic features of severe and lethal cases of COVID-19 in humans, such as diffuse alveolar damage (DAD) with hyaline membrane formation and multi-organ failure associated with hypercoagulability and widespread microvascular fibrin thrombi [57]. Furthermore, lethal COVID-19 in humans has not been attributable to severe neurodissemination.

To better understand the pathogenesis of SARS-CoV-2, systematic characterization of pre-clinical animal models is essential to communicate their translational relevance [20]. Despite extensive use, several aspects of the K18-hACE2 murine model have remained unknown prior to this work, including the spatiotemporal dynamics and pathologic determinants of neuroinvasion and lethal neurodissemination in the context of ACE2 protein and *hACE2* mRNA expression. Our findings demonstrate that lethality of this murine model is associated with neuroinvasion via the ONE with severe neurodissemination irrespective of the inoculation dose (10^4^ vs. 10^6^ PFU). Furthermore, although the portal of entry appears to occur via ACE2 expressing olfactory neuroepithelium, SARS-CoV-2 tropism is not solely restricted to ACE2-expressing cells in K18-hACE2 mice. Thus, the lethal neuropathogenic potential of SARS-CoV-2 in this model is in part dependent on other currently unknown host factors and interaction with viral virulence determinants.

Herein, we utilized a large cohort of K18-hACE2 mice enrolled in either a 14-day natural history or serial euthanasia study to sequentially evaluate SARS-CoV-2 tropism and pathological alterations, spatial and temporal analysis of host factors including inflammatory response and ACE2/*hACE2* expression, and several clinical indices. Survival curve analysis demonstrated that lethality in infected mice only occurs at or after 6 dpi, and in most mice, coincided with the initiation of neurologic signs, neuronal cytopathic effect, and abundance of viral S protein, RNA, and infectious viral particles in the CNS independently of the viral dose used. These observations indicate neuroinvasion and dissemination are a key determinant in the fatal outcome affiliated with this model. Our study also demonstrates that SARS-CoV-2 has a tropism for neurons within the spinal cord (predominantly within the cervicothoracic segments) and retinal ganglion cells, albeit to a lesser degree and only observed in terminal stages of disease. Concurrent brain and spinal cord disease rationalize the neurologic signs observed with this model, which included ataxia, tremors, decreased mobility/responsiveness and decreased urine voiding characterized by severe urinary bladder distention observed at necropsy. The latter is potentially attributed to altered spinal reflexes and/or decreased intervention of the detrusor muscle, which is required for normal micturition. An additional striking clinical feature in infected K18-hACE2 mice at terminal disease was profound hypothermia, which is likely a consequence of dysfunctional hypothalamic control (thermoregulation zone) and generalized neuronal dysfunction associated with SARS-CoV-2 neurotropism. Our results unequivocally demonstrate that neuroinvasion and subsequent neurodissemination are the primary determinants of fatality in this animal model compared to others such as Syrian hamsters, which display more severe pulmonary disease and infection of the ONE but lack evidence of neuroinvasion [58]. Furthermore, Syrian hamsters invariably recover within 14 days following intranasal infection with SARS-CoV-2, with mild residual histopathologic findings that are primarily reflective of repair [22,27,58,59,60]. Very few infected K18-hACE2 mice (2/30) from our high dose survival curve study (14 dpi) survived to 14 dpi and, while residual pulmonary inflammation was observed, these animals did not exhibit any evidence of neuroinvasion. This included normal histologic appearance of CNS with absence of detectable SARS-CoV-2 protein or RNA. Uniquely, both survivors developed pulmonary interstitial aggregates of B lymphocytes which were not observed at earlier time points, suggestive of the development of an adaptive humoral response, which was further supported by the presence of neutralizing antibodies in these two animals. Overall, these findings are of importance to researchers with a particular interest in studying SARS-CoV-2-associated neuropathogenesis, as premature euthanasia due to other clinical features (i.e., weight loss, ruffled fur, and/or respiratory distress) have the potential to precede CNS disease. Such terminal endpoints, if elected, may preclude evaluation of the effects of SARS-CoV-2 in the CNS. Instead, decreased responsiveness/mobility, tremors, ataxia, and hypothermia should be interpreted to reflect neuroinvasion and neurodissemination more accurately and objectively.

To date, the precise mechanism(s) enabling neuroinvasion in the K18-hACE2 model is poorly understood [11,13,15,16,58]. Here, we determined that K18-hACE2 transgenic mice show a significant upregulation in the expression of ACE2 in the nasal cavity compared to wild-type C57BL/6J mice, in which ACE2 expression is undetectable by IHC. This difference between K18-hACE2 and C57BL/6J mice is consistent with the expression of the *hACE2* transgene and is likely a key feature to the neuropathogenesis of this model. Interestingly, temporal analysis of SARS-CoV-2 S protein and RNA in the ONE of transgenic mice preceded and/or occurred simultaneously with infection of neurons within the glomerular and mitral layers of the olfactory bulb, supporting olfactory nerve axonal transport through the cribriform plate as a primary portal of entry. Expression of *hACE2* within neurons in the CNS is overall low and does not directly correlate with our immunohistochemical findings, where ACE2 protein was restricted to blood vessels, ependymal and choroid epithelium with sparing of neurons and their processes. These findings suggest the ACE2 expression in these anatomical compartments could be attributed to *mACE2* and/or indicative of a post-transcriptional event that could be limiting neuronal expression of *hACE2*. These findings suggest that overexpression of hACE2 at the interface of the ONE and olfactory neuronal synapses may be sufficient for initial neuroinvasion, with subsequent neurodissemination mediated by other unknown host mechanisms independent of ACE2.

Infection of brain organoids has been shown to be inhibited using anti-ACE2 antibodies [43]. However, brain organoids do not recapitulate the complex heterogeneity of the CNS, and axonal transport of viral particles into the CNS can hardly be modeled in vitro. Altogether, this suggests that while ACE2 is assuredly an important mediator of CNS neuroinvasion, studying mechanisms of SARS-CoV-2 neurodissemination will likely require the use of complex experimental systems. Neuropilin-1, a transmembrane glycoprotein serving as cell surface receptor for semaphorins and other ligands, as well as Tetraspanin 8 (TSPAN8), have recently been proposed as alternative host receptors for SARS-CoV-2 entry [61,62]. In K18-hACE2 mice neuropilin-1 was sporadically expressed in the ONE, highly expressed in extra-calvarial olfactory nerve fibers, and sporadically in glial cells, the leptomeninges, recruited leukocytes, and blood vessels (Appendix A). Acknowledging SARS-CoV-2 exhibited exclusive neuronal tropism this suggest neuropilin-1 could theoretically have played a contributory role in initial neuroinvasion, but not subsequently in neuron-to-neuron neurodissemination.

Anosmia and ageusia (loss of smell and taste, respectively) represent the earliest and most common but transient neurologic symptoms in people with COVID-19, being reported in ≥50% of cases [12,13,17]. Hyposmia or anosmia has also been clearly characterized in K18-hACE2 mice, occurring between 2–3 dpi, which was characterized through a series of unique behavioral tests requiring a normal sense of smell [36]. Other neurologic manifestations of COVID-19 have been attributed to acute cerebrovascular disease, with cohort studies reporting strokes in 2–6% of hospitalized patients [7,13]. Long-term neurologic sequelae associated with COVID-19 or its effect on neurodegenerative diseases remain unclear [7]. Very little is known about the pathogenesis of these neurologic manifestations and whether they are directly or indirectly associated with SARS-CoV-2. ACE2 expression has been described in humans both in health and with chronic rhinosinusitis, with expression noted in sustentacular cells of the ONE, but not within immature and mature olfactory neurons [63]. This observation led the authors to suggest that anosmia in COVID-19 is likely attributable to an indirect effect of SARS-CoV-2 infection. However, recent studies evaluating the brain and nasal autopsies from patients who died of COVID-19, detected SARS-CoV-2 protein and RNA in cells of neural origin within the ONE and cortical neurons occasionally associated with locally ischemic regions [43,64]. In contrast, a more recent study has determined that, similarly to what was described in Syrian hamsters previously, sustentacular cells (non-neuronal) are the main target cell type in the olfactory mucosa of COVID-19 patients with no evidence of infection of olfactory sensory neurons or olfactory bulb neuroparenchyma [58,65]. Thus, these studies partially support the use of the K18-hACE2 murine model as one with translational significance, even though ischemic lesions have not been reported including results from our study. Even though SARS-CoV-2 infects sustentacular cells within the neuroepithelium of Syrian hamsters [58], the K18-hACE2 and transgenic mice expressing hACE2 under the HFH4 promoter are the only published models that consistently develop neuroinvasion with wild-type virus and, thus, will be particularly useful for studying SARS-CoV-2 neuropathogenesis, particularly the mechanisms of viral neurodissemination within the CNS after initial neuroinvasion via the ONE [33].

Another important observation of the K18-hACE2 model is that SARS-CoV-2 tropism extensively involves infection of ACE2 and *hACE2* negative cells, including certain population of neurons and the vast majority of AT1 pneumocytes. Similarly, sole expression of *hACE2* in some cell types (i.e., CNS blood vessels and bronchiole epithelial cells) does not render these cells susceptible to SARS-CoV-2 even following intranasal exposure and underscores the notion that other undetermined host factors are likely required to allow viral entry. Therefore, this model is relevant for investigating the role of alternative ACE2-independent entry mechanisms.

In conclusion, this study provides a comprehensive spatiotemporal analysis of SARS-CoV-2 infection in the K18-hACE2 transgenic murine model along with an analysis of the contribution of ACE2 in the permissiveness of the model. Our work provides extensive evidence that SARS-CoV-2 can exhibit a marked neurotropism that is associated with lethality, and that this process likely occurs through mechanisms that are in part hACE2-independent. Although we documented significant reactive microgliosis and astrogliosis in terminal neurodissemination, the exact role and molecular determinants of these observations, and their role in neuronal injury of the K18-hACE2 model warrants further research; however, recent work has shown depletion of microglia did not restrict SARS-CoV-2 replication [66]. Lethal CNS invasion, combined with the absence of severe pulmonary hallmarks associated with lethal COVID-19, therefore calls for attentive caution when utilizing the K18-hACE2 mouse model to investigate certain aspects of SARS-CoV-2 pulmonary pathogenesis. Furthermore, due to the acute and fulminant neuroinvasion and dissemination, the protective ability of certain anti-viral therapies, and T-cell based vaccines against lethal challenge in this model might indeed be underestimated, which is reflected in several studies that have utilized terminal timepoints preceding neuroinvasion as their efficacy endpoints [67,68,69]. Regardless, the K18-hACE2 mouse model represents a promising model for understanding the mechanisms governing SARS-CoV-2 neuroinvasion, neurodissemination, and evaluating potent and fast-acting prophylactic countermeasures. Lastly, this model may serve useful in evaluating efficacy of therapeutics to block development of reactive/injurious microglial and/or astrocyte phenotypes if determined to play a key role in the neuronal injury observed in this model.

## Figures and Tables

**Figure 1 viruses-14-00535-f001:**
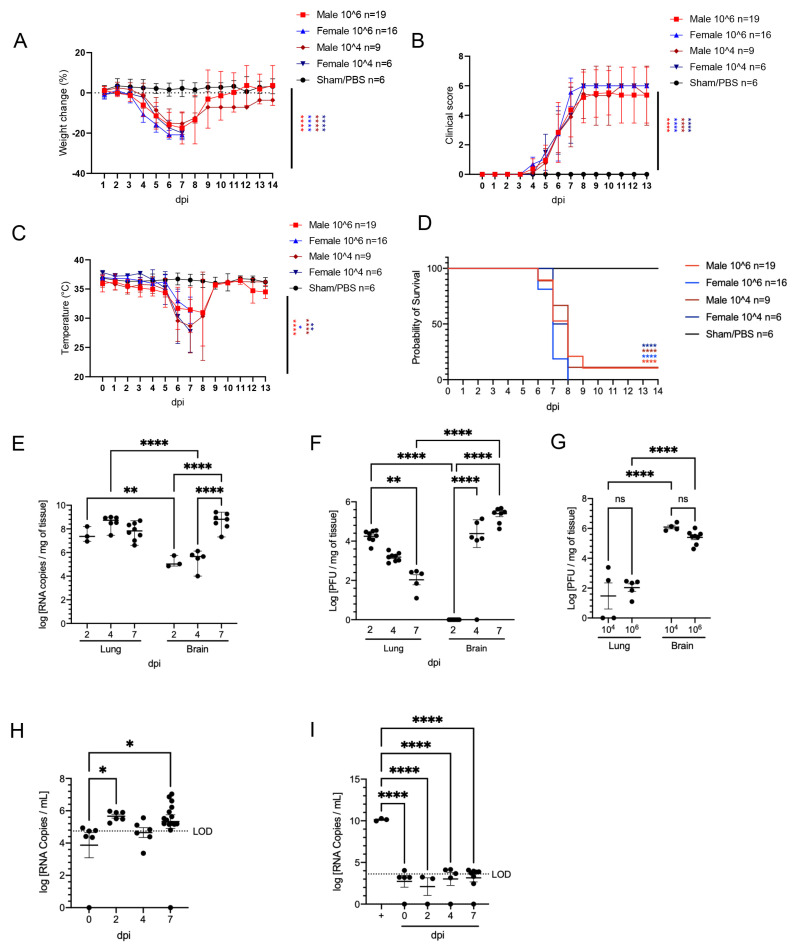
SARS-CoV-2 caused lethal disease in K18-hACE2 mice irrespective of dose (10^4^ vs. 10^6^ PFU). K18-hACE2 mice (n = 50) were inoculated intranasally with either 1 × 10^4^ or 1 × 10^6^ plaque forming units (PFU), with n = 6 additional Sham/PBS negative controls. Body weight (**A**), clinical signs (**B**), body temperature (**C**), and survival (**D**) were monitored daily in Sham/PBS animals (black) and in infected animals (male, red; female, blue; up to 14 dpi). Mice meeting euthanasia criteria were counted dead the following day. Viral loads (viral RNA genome copy numbers/mg of tissue) or viral titers (infectious virus particles; PFU/mg of tissue) were quantified in the lung and brain (**E**–**G**). RNA copies were also examined in the serum (genome copies/mL) either directly on serum (**H**) or via a re-infectivity assay (**I**) using Vero E6 cells. The limit of detection is shown with a dashed line. Clinical data (**A**–**D**): 10^4^ PFU; male (n = 9), female (n = 6); 10^6^ PFU male (19), female (n = 16); Sham/PBS male (n = 3), female (n = 3). Molecular and virologic data (**E**–**I**). 10^6^ RT-PCR: lung 2 dpi (n = 3), 4 dpi (n = 6), 7 dpi (n = 8); brain 2 dpi (n = 3), 4 dpi (n = 5), 7 dpi (n = 7). 10^6^ PFU analysis: lung 2 dpi (n = 8), 4 dpi (n = 8), 7 dpi (n = 5); brain 2 dpi (n = 8), 4 dpi (n = 8), 7 dpi (n = 8). 10^4^ PFU analysis: lung 7 dpi (n = 4); brain 7 dpi (n = 4).10^6^ Serum RT-PCR assay: Sham (n = 6), 2dpi (n = 6), 4 dpi (n = 6), 7 dpi (n = 15). 10^6^ Serum infectivity assay: Sham (n = 5), 2dpi (n = 3), 4 dpi (n = 5), 7 dpi (n = 8). One-way or two-way ANOVA. * *p* ≤ 0.05, ** *p* ≤ 0.01, **** *p* ≤ 0.0001. For A–D, shades of blue and red asterisks compare sham group vs. male group, and sham group vs. female group, respectively, with darker shades designated for 10^4^ PFU, and brighter shades for 10^6^ PFU.

**Figure 2 viruses-14-00535-f002:**
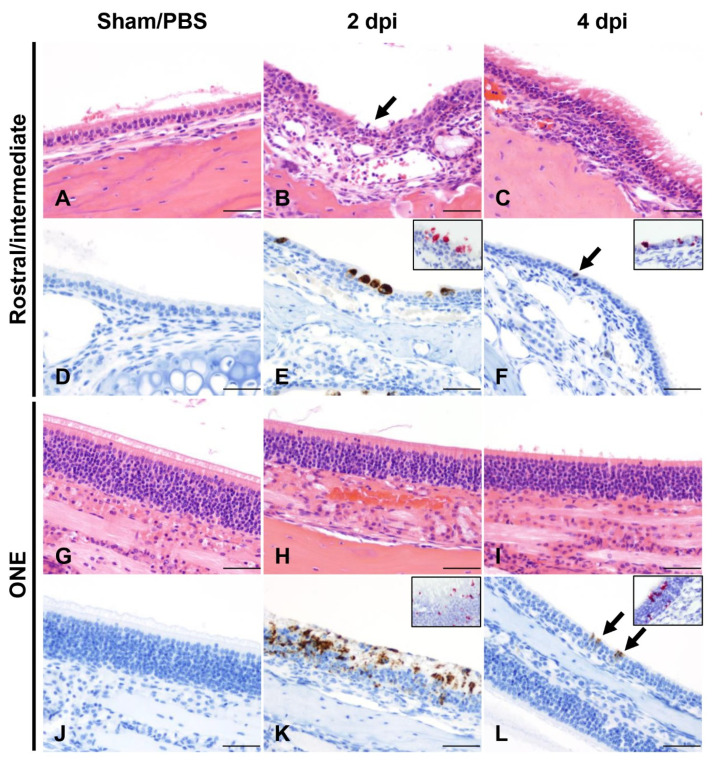
Temporal analysis of SARS-CoV-2 infection in the nasal cavity of K18-hACE2 mice. Histological changes, and viral protein (brown) and RNA (red) distribution and abundance were assessed in non-infected (Sham/PBS: **A**,**D**,**G**,**J**; n = 3) and infected mice at 2 (**B**,**E**,**H**,**K**, n = 3) and 4 (**C**,**F**,**I**,**L**, n = 5) days following intranasal inoculation. At 2 dpi, neutrophilic rhinitis in the rostral and intermediate turbinates (**B**, arrow) correlated with intraepithelial SARS-CoV-2 protein and RNA (**E**, inset). Viral protein and RNA were detected in the olfactory neuroepithelium (ONE, **K** and inset) in the absence of histologic lesions (**H**). At 4 dpi, only sporadically infected cells were noted in the epithelium lining the nasal turbinates and ONE (**F**,**L**, arrow, and insets) in the absence of histologic lesions (**C**,**I**). Sham/PBS-infected are depicted in **A**, **D**, **G**,**J**. H&E, DAB IHC (viral protein), and Fast Red ISH (viral RNA), 200× total magnification. Bar = 100 μm.

**Figure 3 viruses-14-00535-f003:**
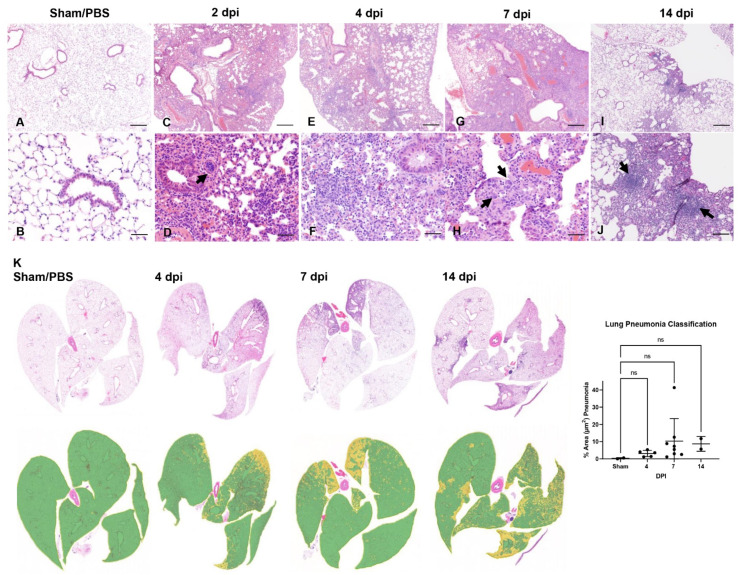
Temporal qualitative and quantitative analysis of SARS-CoV-2 pneumonia in K18-hACE2 mice. Lung tissues from non-infected mice (PBS/Sham inoculated: (**A**,**B**) and from infected mice at 2 dpi (**C**,**D**), 4 dpi (**E**,**F**), 7 dpi (**G**,**H**) and 14 dpi (n = 2; **I**,**J**) following intranasal inoculation were analyzed. Subgross histological images of the lungs and corresponding pneumonia classifiers for each timepoint are depicted in panel (**K**) (green = normal; yellow = pneumonia). Mild-to-moderate interstitial pneumonia was evident starting at 2 dpi with frequently reactive blood vessels (**D**, arrow). At 7 dpi, alveolar type 2 (AT2) cell hyperplasia was observed (**H**, arrows). Residual mild-to-moderate pneumonia was observed in the two male survivors at 14 dpi from the survival curve, with rare sporadic lymphoid aggregates (**J**-arrows). H&E, 50× (**A**,**C**,**E**,**G**,**I**; bar = 500 μm), 200× (**B**,**D**,**F**,**H**,**J**; bar = 100 μm) and 1× (**K**) total magnification. Pneumonia classifier: PBS/Sham (n = 2), 4 dpi (n = 5), 7 dpi (n = 8), 14 dpi (n = 2). One-way ANOVA; ns, non-significant.

**Figure 4 viruses-14-00535-f004:**
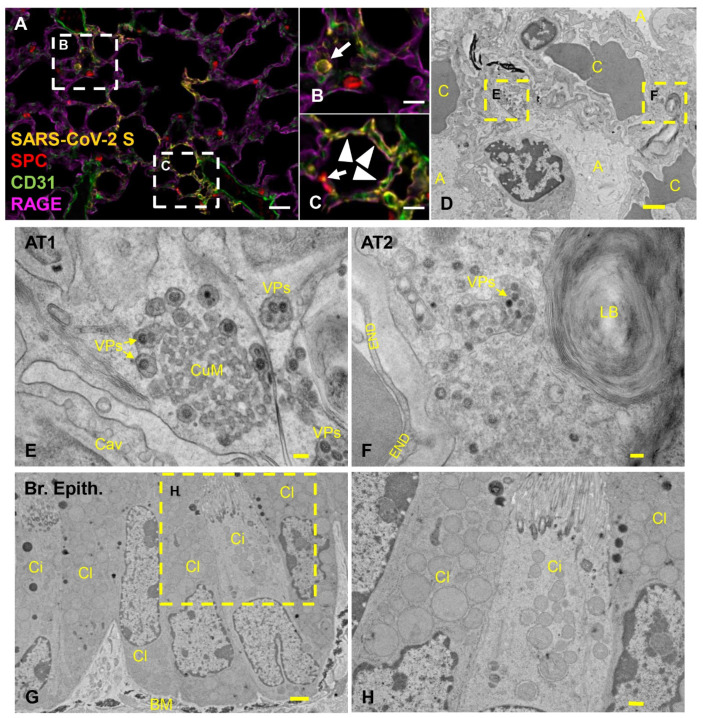
SARS-CoV-2 tropism following intranasal inoculation in K18-hACE2. (**A**–**C**) At 4 dpi, SARS-CoV-2 (yellow) tropism for RAGE+ alveolar type 1 (AT1, magenta) and scattered SPC+ alveolar type 2 (AT2, red) pneumocytes (**B**,**C**, arrowheads, and arrows, respectively) but not for CD31+ endothelial cells. (**B**,**C**) represent higher magnification images of the white hashed boxes from (**A**), respectively. At 6 dpi-terminal disease (**D**–**F**), interpretation of transmission electron microscopy images illustrated virus particles (VPs) bound by vesicles in AT1 (**E**) and AT2 (**F**) cells. AT1 contained abundant caveolae. Another unique feature observed in AT1 cells was the presence of cubic membranes (CuM). AT2 pneumocytes were characterized by presence of lamellar bodies (LB). (**E**,**F**) represent higher magnification images of the yellow hashed boxes from (**D**). (**G**,**H**) Viral particles or viral induced membrane alterations were not identified in ciliated or non-ciliated club (Cl) bronchiolar epithelial cells. (**H**) represents a higher magnification of the yellow hashed box in (**G**). Multiplex fluorescent IHC, 100× (**A**; bar = 100 μm) and 200× (**B**,**C**; bar = 50 μm) total magnification. TEM, bar = 2 μm (**D**), 100 nm (**E**–**H**), and 3 μm (**G**). A, alveolar lumen; BM, basement membrane; C, capillary; Cav, caveolae; Ci, ciliated epithelium; Cl, club epithelium; CuM, cubic membranes; DMVs, double-membrane vesicles; END, endothelium; VPs, viral particles.

**Figure 5 viruses-14-00535-f005:**
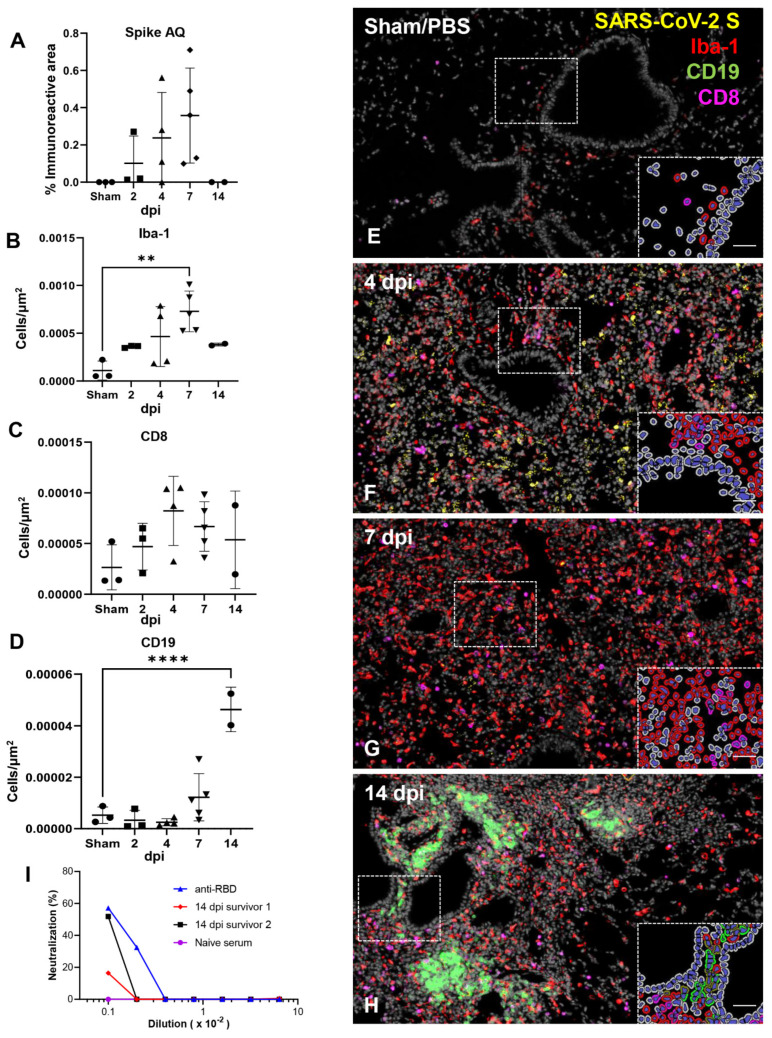
Temporal immunoprofiling of the pulmonary host inflammatory response to SARS-CoV-2. (**A**–**H**) Quantification and 4-plex fluorescent IHC targeting SARS-CoV-2 Spike (**A**,**E**–**H**), and macrophage Iba-1+ (**B**,**E**–**H**), CD8+ (**C**,**E**–**H**) and CD19+ cell (**D**,**E**–**H**) infiltration in the lung of PBS/Sham inoculated mice and in SARS-CoV-2 inoculated mice (2 dpi (n = 3), 4 dpi (n = 4), 7 dpi (n = 5) and 14 dpi (n = 2). In inoculated mice, SARS-CoV-2 Spike peaked between 4–7 dpi (**A**,**F**,**G**). Iba-1+ macrophages (red) increased significantly peaking at 7 dpi (**B**,**G**), along with a lower infiltration of CD8+ T lymphocytes-magenta that peaked between 4–7 dpi (**C**,**F**,**G**) while Sham/PBS mice had low residual inflammatory cells (**E**). CD19+ B cells arranged in aggregates were only evident in the two male survivors euthanized at 14 dpi (**D**,**H**). Insets depict immune cell phenotyping outputs that were applied across the whole slide image. Sham/PBS-infected mice (n = 3) were used as baseline controls for quantitative analysis. Multiplex fluorescent IHC (**E**–**H**): 100× and 400× (insets) total magnification, Bar = 100μm. (**I**) Neutralizing activity of serum isolated from a naïve/non-infected K18-hACE2 (purple) and from the two male14 dpi survivors (survivor 1 and 2, red and black, respectively). An anti-SARS-CoV-2 Spike RBD antibody (anti-RBD, blue) was used as a positive control. Serum was serially diluted by 2-fold. One-way ANOVA. ** *p* ≤ 0.01; **** *p* ≤ 0.0001.

**Figure 6 viruses-14-00535-f006:**
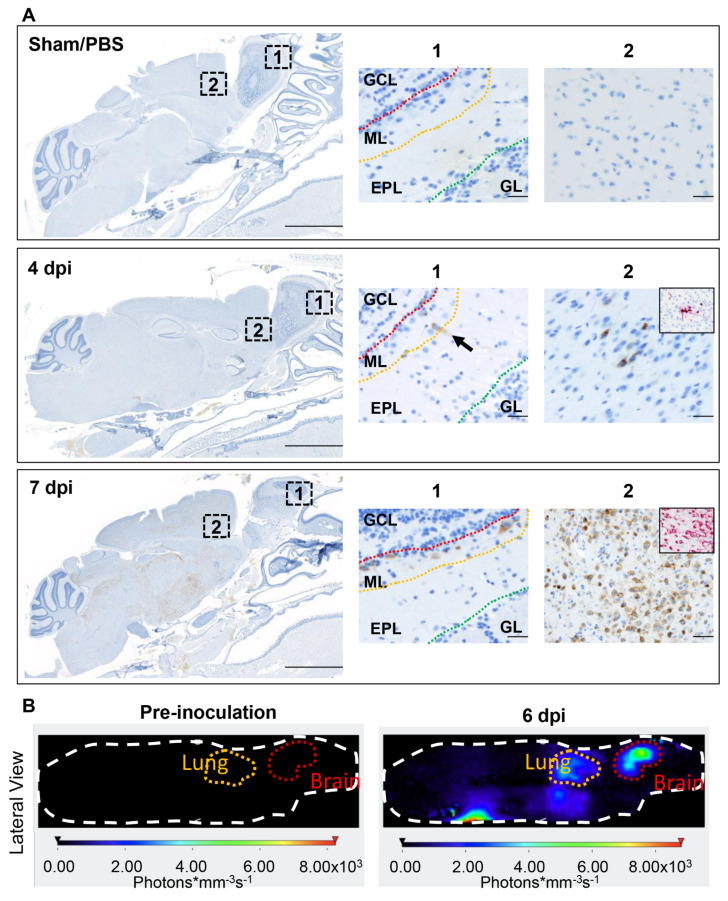
Invasion of SARS-CoV-2 into the central nervous system. (**A**) Sagittal sections of the head of non-infected (Sham/PBS, top panel) and infected (4 and 7 dpi, middle and bottom panel, respectively) were analyzed for viral protein and RNA distribution. At 4 dpi (middle panel), SARS-CoV-2 infected neurons within the mitral layer of the olfactory bulb (1, arrow) as well as small clusters of neuronal bodies within the cerebral cortex (2, SARS-CoV-2 RNA in inset). At 7 dpi (bottom panel), SARS-CoV-2 protein was widespread along the mitral layer of the olfactory bulb (1) and throughout the central nervous system (2, SARS-CoV-2 RNA in inset) with exception of the cerebellum. EPL, external plexiform layer; GCL, granular cell layer; GL, glomerular layer; ML, mitral layer. DAB (viral protein) and Fast Red (viral RNA). 7.5× (bar = 2.5 mm) and 200× (bar = 100 μm) total magnification. On the right of each panel, pictures labelled 1 and 2 are 266× total magnification insets represented by the hashed squares labeled in the lower (7.5×) magnification images. (**B**) Representative three-dimensional profile view (right side) of a K18-hACE2 mouse following inoculation with a SARS-CoV-2 NL virus (10^6^ PFU). NanoLuc bioluminescent signal was detected and quantified at 6 dpi following fluorofurimazine injection (Sub-cutaneous) using the InVivoPLOT (InVivoAx) system and an IVIS Spectrum (PerkinElmer) optical imaging instrument. Location of the lungs and brain are indicated.

**Figure 7 viruses-14-00535-f007:**
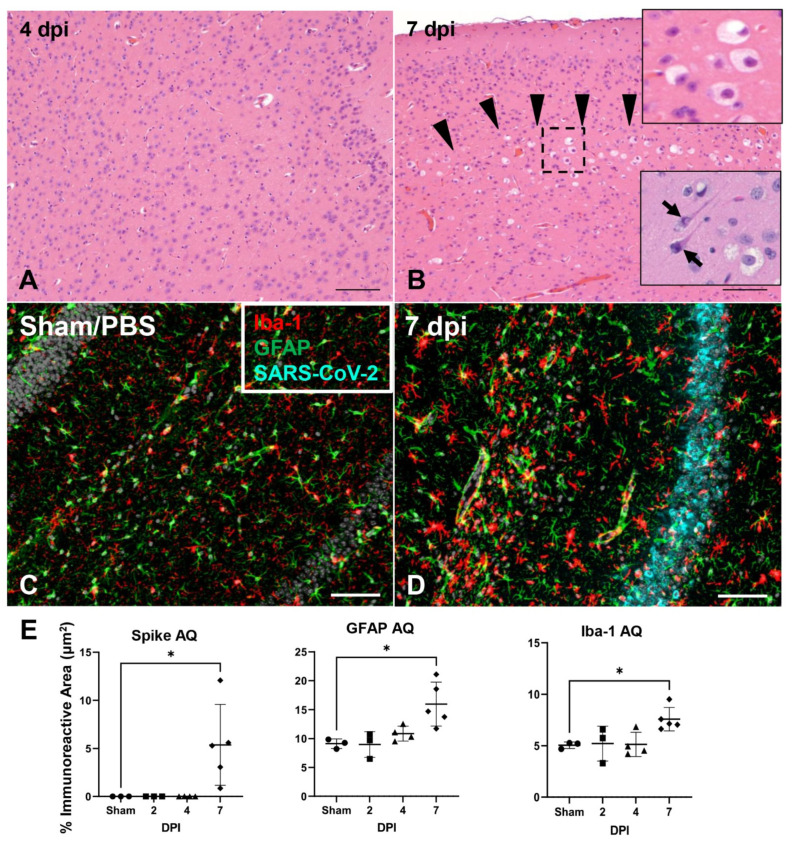
SARS-CoV-2-associated neuronal morphological changes, neuronal antigen abundance and glial response. Morphological changes in the brain were noted as early as 6 dpi (**A**,**B**) and were characterized by variable spongiosis (**B**, arrowheads, and top inset) with neuronal degeneration and necrosis (**B**, bottom inset and arrows) involving multiple areas within the cerebral cortex and elsewhere. (**C**–**E**) Quantification of 3-plex fluorescent IHC targeting SARS-CoV-2 Spike protein, astrocytes (GFAP) and microglia (Iba-1) in the brain of Sham/PBS mice and in inoculated mice (2, 4, 7 and 14 dpi). The amount of viral protein rapidly and markedly increased by 7 dpi, along with an intense astrocytic and microglial response (**C**–**E**). H&E, 100×, bar = 200 μm. Multiplex IHC, 200× total magnification, bar = 100 μm. One-way ANOVA; * *p* ≤ 0.05.

**Figure 8 viruses-14-00535-f008:**
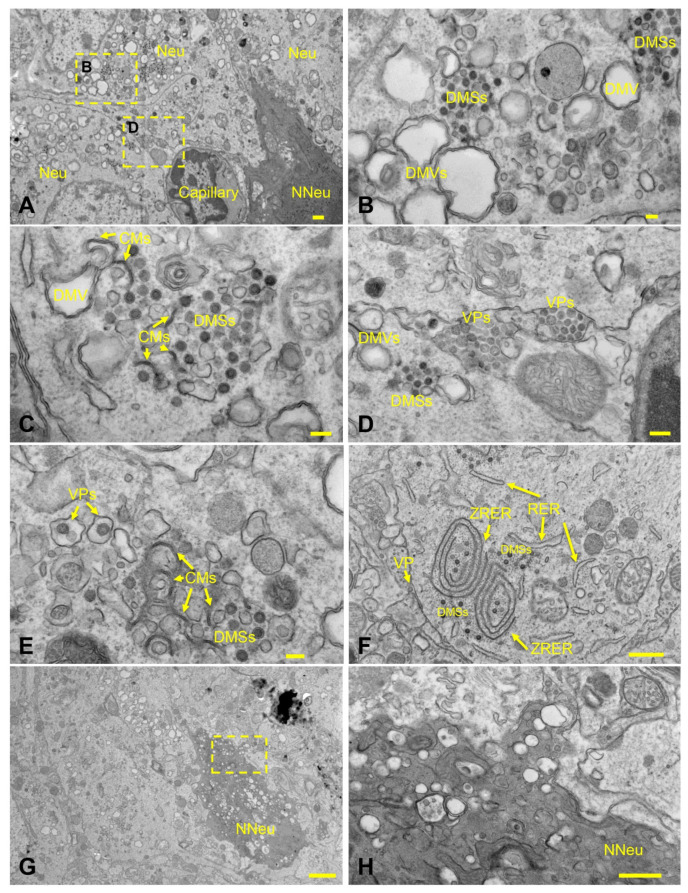
Ultrastructural features of SARS-CoV-2-infected neurons. (**A**) Neurons (Neu) contain abundant intracytoplasmic viral particles and various replication-associated intracytoplasmic membranous structures. Necrotic neurons are diffusely electron dense (NNeu). (**B**) Higher magnification of the squared area on (**A**) depicting double-membrane vesicles (DMVs) and spherules (DMSs). (**C**) Cubic membranes (CMs) are also noted among DMVs and DMSs. (**D**) Higher magnification of the squared area on (**A**). Mature viral particles are indicated as VPs. (**E**,**F**) VPs are associated with membranous structures related to viral replication (DMVs, DMSs and CMs) and with the rough endoplasmic reticulum (RER). (**G**) Necrotic neurons (NNeu) are diffusely electron dense and contain numerous cytoplasmic vacuoles. (**H**) Higher magnification of squared area on (**G**). Note the high cytoplasmic electron-density. Scale bars = 100 nm.

**Figure 9 viruses-14-00535-f009:**
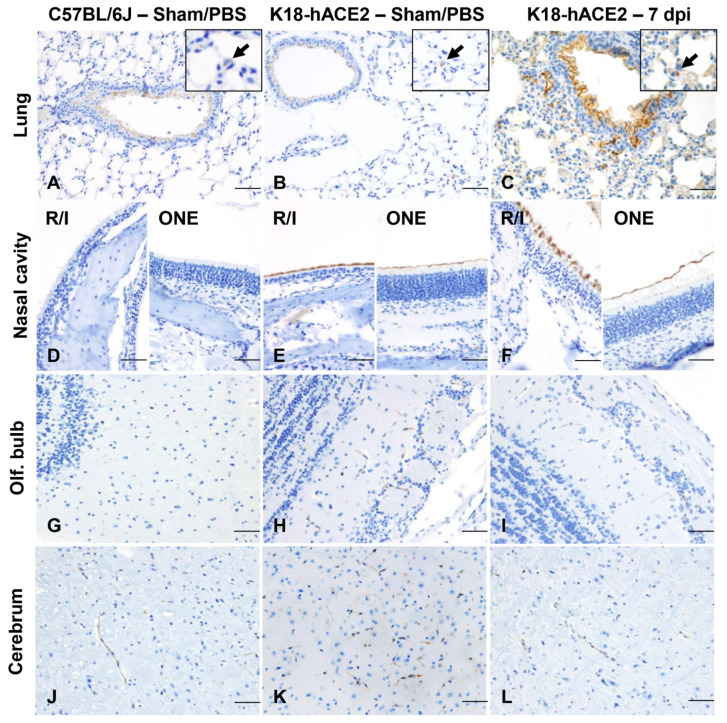
Distribution of ACE2 in lungs, nasal cavity, brain, and olfactory bulb of wild-type C57BL/6J and uninfected and SARS-CoV-2 infected K18-hACE2 mice. Lung (**A**–**C**), nasal (rostral/intermediate turbinates [R/I]) and olfactory epithelium (ONE) (**D**–**F**), olfactory bulb (**G**–**I**) and brain (**J**–**L**) from non-infected C57BL/6J, and from non-infected and infected K18-hACE2 mice (7 dpi). K18-hACE2 mice were analyzed via immunohistochemistry using a cross-reactive anti-ACE2 antibody. In the lungs (**A**–**C**), ACE2 expression (brown) was mostly restricted to the apical membrane of bronchiolar epithelial cells with scattered positive AT2 cells (inset arrows). Nasal (rostral/intermediate turbinates [R/I]) and olfactory epithelium (ONE) were devoid of ACE2 in C57BL/6J mice (**D**), but expression was enhanced in K18-hACE2 mice with intense apical expression (**E**,**F**). ACE2 expression within the olfactory bulb (**G**–**I**) and the brain (**J**–**L**) was restricted to capillary endothelium with no neuronal expression. DAB, 200× total magnification. Bar = 100 μm.

**Figure 10 viruses-14-00535-f010:**
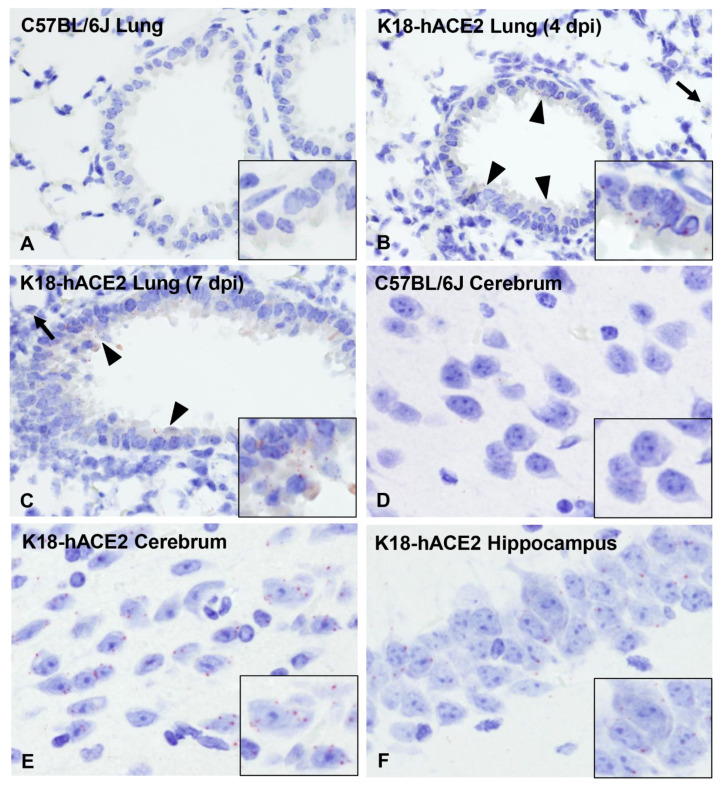
Expression and distribution of *hACE2* mRNA in the brain and lungs of non-transgenic wild-type C57BL/6J and K18-hACE2 transgenic mice via RNAscope^®^ ISH. (**A**–**C**) *hACE2* lung expression. While no expression of *hACE2* was noted in the lungs of wild-type C57BL/6J mice (**A**), *hACE2* was expressed in the bronchiolar epithelium (arrowheads) and sporadic AT2 cells (arrows) in transgenic K18-hACE2 mice (**B**,**C**), which correlated with immunohistochemical findings. (**D**–**F**) *hACE2* brain expression. *hACE2* was not expressed in the Cerebrum of C57BL/6J mice (**D**) but in clusters of neurons within the cerebrum (**E**) and hippocampus (**F**). Fast Red, 400× total magnification. Bar = 50 μm.

**Table 1 viruses-14-00535-t001:** Clinical scoring system used for clinical monitoring of SARS-CoV-2-infected K18-hACE2 mice.

Category	Score = Criteria
Body weight	1 = 10–19% loss
Respiration	1 = rapid, shallow, increased effort
Appearance	1 = ruffled fur, hunched posture
Responsiveness	1 = low to moderate unresponsiveness
Neurologic signs	1 = tremors

**Table 2 viruses-14-00535-t002:** Antibodies and antigen retrieval conditions for the assays performed in this study.

	Sequence	Antigen Target	Species Origin	Clone	Manufacturer	Catalog	Primary Antibody Dilution	Antigen Retrieval (Ventana)	Chromogen or Fluorophore
Assay 1	NA	SARS-CoV-2 Spike (S)	Mouse	E7U6O	Cell Signaling Technology (Danvers, MA, USA)	Pre-commercialization	1:1000	CC1 (Tris)	DAB
Assay 2	NA	Angiotensin converting enzyme 2 (ACE2)	Rabbit	EPR34435	Abcam (Waltham, MA, USA)	ab108252	1:200	CC1 (Tris)	Discovery Red and DAB
Assay 3	NA	Neuropilin-1	Rb	EPR3113	Abcam	ab81321	1:50	CC2 (Citrate)	DAB
Assay 4	1	SARS-CoV-2 S	Mouse	E7U6O	Cell Signaling Technology	Pre-commercialization	1:1000	CC1 (Tris)	Opal 480
	2	Iba-1	Rabbit	Polyclonal	WAKO (Richmond, VA, USA)	019-19741	1:2000	CC2 (Citrate)	Opal 570
	3	GFAP	Rabbit	Polyclonal	DAKO (Glostrup, Denmark)	Z0334	1:500	CC1 (Tris)	Opal 690
Assay 5	1	CD8	Rabbit	D4W2Z	Cell Signaling Technology	98941	1:200	CC1 (Tris)	Opal 620
	2	SARS-CoV-2 S	Mouse	E7U6O	Cell Signaling Technology	Pre-commercialization	1:1000	CC1 (Tris)	Opal 570
	3	CD19	Rabbit	D4V4B	Cell Signaling Technology	90176	1:600	CC2 (Citrate)	Opal 520
	4	Iba-1	Rabbit	Polyclonal	WAKO	019-19741	1:2000	CC2 (Citrate)	Opal 690
Assay 6	1	RAGE	Rat	EPR21171	R&D (Minneapolis, MN, USA)	MAB1179q-100	1:50	CC1 (Tris)	Opal 480
	2	SARS-CoV N	Rabbit	Polyclonal	Novus biologicals (Littleton, CO, USA)	NB100-56576	1:200	CC1 (Tris)	Opal 570
	3	Prosurfactant C Protein	Rabbit	Polyclonal	Seven Hills Bioreagents (Cincinnati, Ohio, USA)	WRAB-9337	1:800	CC2 (Citrate)	Opal 690
	4	CD31	Rabbit	D8V9E	Cell Signaling Technology	77699S	1:100	CC2 (Citrate)	Opal 520

**Table 3 viruses-14-00535-t003:** SARS-CoV-2 viral protein abundance in tissues derived from SARS-CoV-2-infected K18-hACE2 mice. Median scores are represented along with ranges between brackets when applicable.

DPI	AT1/AT2	Bronchioles	Rostral Turbinates	Intermediate Turbinates	ONE	Olf. Bulb	Brain	Spinal Cord (CT)	Spinal Cord (LS)	GI *	Kidneys
PBS/Sham (n = 3)	0	0	0	0	0	0	0	0	0	0	0
2 (n = 3)	2 (1–2)	0	1 (0–2)	2 (1–2)	1 (1–2)	0	0	0	0	0	0
4 (n = 5)	2 (1–3)	0	0 (0–1)	0 (0–1)	1 (0–1)	0 (0–1)	0 (0–1)	0	0	0	0
6–8 (n = 13)	2 (1–3)	0	0	0	1 (0–1)	1 (0–2)	3 (0–3)	1 (0–2)	0 (0–1)	0	0
14 (n = 2)	0	0	0	0	0	0	0	0	0	0	0

0, no SARS-CoV-2 protein observed; 1, 0 to 5% of cells within a high magnification (400×) field are positive for viral protein; 2, 5 to 25% of cells within a high magnification (400×) field are positive for viral protein; 3, >25 to <50% of cells within a high magnification (400×) field are positive for viral protein. NA, not available. AT1, alveolar type 1 pneumocytes; AT2, alveolar type 2 pneumocytes; ONE, olfactory neuroepithelium; CT, cervicothoracic segment; LS, lumbosacral segment; GI, gastrointestinal tract. * Sections examined included stomach, small intestine (duodenum, jejunum, and ileum) and large intestine (cecum and colon).

## Data Availability

The data presented in this study are available on request from the corresponding authors.

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
