# Peer review of "Fatal Neurodissemination and SARS-CoV-2 Tropism in K18-hACE2 Mice Is Only Partially Dependent on hACE2 Expression"

_viruses, 2022, doi:10.3390/v14030535_

Round 1
Reviewer 1 Report
In this manuscript “Fatal neurodissemination and SARS-CoV-2 tropism in K18-hACE2 mice is only partially dependent on hACE2 expression” by Carossino et al., the authors assess the pathogenicity of SARS-CoV-2 in the previously developed and validated K18 hACE2 transgenic mouse model of SARS-CoV-2 infection and associated COVID-19 disease using low and high viral doses. Authors demonstrate that SARS-CoV-2 virulence in this mouse model was mediated by its ability to infect the brain via the olfactory neuroepithelium.
There is a need for animal models to interrogate SARS-CoV-2 infection, tropism, virulence, and transmission; and for the testing of prophylactics and therapeutics to control SARS-CoV-2 infection. Several studies have already demonstrated how K18 hACE2 transgenic mice, although perhaps not ideal, represent an excellent model to study SARS-CoV-2 infection. This manuscript add to the already existing literature on K18 hACE2 mice to study SARS-CoV-2. There are some comments that the authors may want to address:
- Several groups have already demonstrate the feasibility of using K18 hACE2 transgenic mice to study SARS-CoV, and, more recently SARS-CoV-2. Likewise, previous studies have demonstrate that this lethal K18 hACE2 transgenic mouse model is based on the ability of SARS-CoV/SARS-CoV-2, to infect the brain. Moreover, it has been reported that aerosol infection of K18 hACE2 transgenic mice does not result in clinical disease as observed in mice infected intranasally. Finally, previous research has demonstrated that viral titers in the brain peak at the time of euthanasia. Thus, although this manuscript provides some novelty in the cause of SARS-CoV-2 dissemination in the brain, there is limited novelty to that already available in the literature. Perhaps the authors should focus on what is novel in this study compared to previous published data.
- Since infection of K18 hACE2 mice with SARS-CoV-2 at the reported doses results in death between days 6-8 post-infection, it is unclear how animals and how many were used in the different experimental groups. It is also unclear the rational to select dissimilar dpi in different experiments. Also, it has been previously described that morbidity and mortality of SARS-CoV-2 in K18 hACE2 mice is dose dependent. This is different to the results shown in this manuscript.
- It will important to quantify the data in the images shown in different figures.
- Some of the experiments lack the use of important key controls.
- The authors should contemplate decreasing the number of figures and reducing some of the sections in the manuscript (e.g. material and methods, and discussion).
- It has not been possible to accurately assess some of the statements in the manuscript. Several references have not been included in the bibliography section.
- The authors should revise the document for consistency and accuracy. For example, the material and methods section indicates cells and viruses but only cells have been described. Number of cells in the same plate format change between experiments.
- It is unclear if the SARS-CoV-2 used in this study, either the natural isolate(s) or the recombinant expressing Nluc, have been sequenced to confirm that mutations potential affecting virulence have not been introduced during amplification in Vero E6 cells. Based on the information available in the literature on SARS-CoV-2 infection of the K18 hACE2 transgenic mouse model, 10E4 pfu should not be considered low infectious dose.
Author Response
Please see the attachment, thank you kindly for your feedback.

Reviewer 2 Report
Carossino et al studied SARS-CoV-2 neurotropism using K18-hACE2 mice.
I think that the manuscript contains excellent data on neurodissemination of SARS-CoV-2 in mice. Experiment was well designed and performed.
Author Response
Thank you for your positive feedback on our manuscript. We greatly appreciate you taking the time to peer review this work.
Reviewer 3 Report
In this manuscript, Carossino et al. conducted a series of immunohistology assessment on lung, brain and spinal cord of K18-hACE2 transgenic mice infected with WA1/2020 strain bearing 614D - a first SARS-CoV-2 isolate detected in the U.S. Some images from immunohistochemistry staining and TEM are very impressive. However, the major issue is that the study lacks clear focus and is not objective-oriented. Many results/images/experiments are irrelevant or unsupportive, making an undeniable impression that the authors were finishing around and were desperate to make some significant connections among the vast amount of data presented.
If the authors intended to claim viral neurodissemination is partially responsible for WA1/2020 induced lethality in K18-hACE2 mice as the main take-home message, they should delete all irrelevant data/images/experiments (Fig. 2, 3, 4, 5, 6, 11, 14) and just focus on the histology evaluation of brain tissues of WA1/2020 infected K18-hACE2 mice. The manuscript would require significant facelift revisions, remove those inaccurate descriptions or overstatements and keep up with the latest development in the field.
Line 129-132: “several histologic hallmarks of severe COVID-19 were lacking in this model (i.e. lack of diffuse alveolar damage and capillary microthrombi)” and line 581-583 “hyaline membranes, vascular thrombosis, … … “. These statements are inaccurate. The disease pathology is virus dependent. WA1/2020 is relatively weak compared to later viruses that bear D614G substitution. Many of severe COVID-19 symptoms such as diffuse alveolar damage, capillary microthrombi hyaline membrane, etc. are recapitulated in K18-hACE2 mice infected with 614G virus.
Using body weight loss >=20% as the surrogate of death leads to overestimate mortality of WA1 infected K18-hACE2 mice. As the authors demonstrated in the study, WA1 causes viral neutrodissemination in K18-hACE2 mice and infected mice die of severe brain hemorrhage after inoculation of high infectious doses. Thus, neurologic signs should be the major surrogate marker of moribundity instead of 20% weight loss. How did the authors measure the core body temperature?
Line 259, 262-263: how is it possible after tissues were preserved in 600 ul of RNAlater and stored in -80C, the authors could still weigh “20-40 mg of tissues” or “sampling was performed on several regions … including the olfactory bulb, cerebral cortex ….”? Why not directly homogenate the whole brain or whole lung?
Line 301 Never heard “Serum infectivity assay”. If serum has infectivity, then how come it contains neutralizing antibody that doesn’t neutralize it (line 309)? Where and when were these serum samples collected from?
Line 447: two male mice survived the high dose infection, and one male mouse survived the low dose infection. They were clearly not properly anesthetized under isoflurane before infection. What is the purpose to include only two of the survivors in analyses (Fig. 3, 4, 6 and 7)? What made these two survivors special (Fig. 6I)? Line 648-651: “minimal inflammatory cells were observed in sham-inoculated K18-hACE2 mice supporting that the two survivors were de-facto infected with SARS-CoV-2, which was further supported by the presence of neutralizing antibodies in their serum as compared to naïve mice”? What did the authors intend to state here? Claiming that these survivors had pre-existing immunity before infection? If so, how? Shouldn’t they be disqualified from the studies if they were accidentally exposed or immunized before?
The authors have overstated many immunohistochemistry observations. e.g., Fig.2E line 492: clearly not “abundant” intraepithelial SARS-CoV-2 spike staining. Similarly, Fig 8A 7dpi 1, absolutely not “widespread” SARS-CoV-2 protein in the mitral layer of the olfactory bulb (line 713). Line 692-693: “neuronal morphologic changes directly correlated with abundant neuronal immunoreactivity for SARS-CoV-2 S protein and viral RNA”. Where are the images/figures that show “abundant neuronal immunoreactivity”? Line 703-705: where are the data to show “SARS-CoV-2 S protein and RNA preceded histological findings with rare detection as early as 4 dpi”? Did the authors have immunohistochemistry staining on 1, 2, or 3 dpi? Line 726-727: where are the images that show “animals with abundant neuronal degeneration and necrosis”?
If “evidence of viral infection was not observed in microglia or astrocytes (line 752-753)”, then what was shown in Fig 9C and why did the authors state “the amount of viral protein rapidly and markedly increased by 7 dpi, along with an intense astrocytic and microglial response (line 761-762)”? Which statement is correct?
What is the purpose of comparing ACE2 expression of K18-hACE2 to C57BL/6J? Is C57BL/6J susceptible to SARS-CoV-2?
Author Response
Thank you for your feedback. Please find attached our responses to your comments.
